# Large-scale Training of Foundation Models for Wearable Biosignals

**Salar Abbaspourazad**[*], **Oussama Elachqar, Andrew C. Miller, Saba Emrani,**
**Udhyakumar Nallasamy, Ian Shapiro**[*]
Apple

## Abstract

Tracking biosignals is crucial for monitoring wellness and preempting the development of severe medical conditions. Today, wearable devices can conveniently record various biosignals, creating the opportunity to monitor health status without disruption to one's daily routine. Despite widespread use of wearable devices and existing digital biomarkers, the absence of curated data with annotated medical labels hinders the development of new biomarkers to measure common health conditions. In fact, medical datasets are usually small in comparison to other domains, which is an obstacle for developing neural network models for biosignals. To address this challenge, we have employed self-supervised learning using the unlabeled sensor data collected under informed consent from the large longitudinal Apple Heart and Movement Study (AHMS) to train foundation models for two common biosignals: photoplethysmography (PPG) and electrocardiogram (ECG) recorded on Apple Watch. We curated PPG and ECG datasets from AHMS that include data from ∼141K participants spanning ∼3 years. Our self-supervised learning framework includes participant level positive pair selection, stochastic augmentation module and a regularized contrastive loss optimized with momentum training, and generalizes well to both PPG and ECG modalities. We show that the pre-trained foundation models readily encode information regarding participants' demographics and health conditions. To the best of our knowledge, this is the first study that builds foundation models using large-scale PPG and ECG data collected via wearable consumer devices – prior works have commonly used smaller-size datasets collected in clinical and experimental settings. We believe PPG and ECG foundation models can enhance future wearable devices by reducing the reliance on labeled data and hold the potential to help the users improve their health.

## 1 Introduction

Recent advances in wearable device technology allow for the recording of various biosignals, which can be used to monitor users' overall wellness. Two of the most commonly collected biosignals from wearable devices are photoplethysmography (PPG) and electrocardiogram (ECG). PPG measures volumetric changes in arterial blood flow and contains a wide variety of biological information, and ECG measures cardiac electrical activity and contains information about cardiac function. While these biosignals hold significant potential, the absence of curated datasets with annotated medical labels has been an obstacle for developing digital biomarkers employing deep neural networks. Medical datasets are typically collected in lengthy and expensive health studies, require domain expertise for annotation, and are usually collected from limited number of participants, which can make the learned models less generalizable to broad, demographically varied populations.

Large-scale training of foundation models via self-supervised learning (SSL) has shown promise in domains such as natural language processing (Devlin et al., 2019; OpenAI, 2023), computer vision (Chen et al., 2020; 2021; Oquab et al., 2023) and speech recognition (Baevski et al., 2020; 2022). Self-supervised learning often does not require explicit labels, making it suitable to pre-train foundation models on unlabeled biosignals, and has been recently proven successful for health applications (Hallgrímsson et al., 2018; Cheng et al., 2020; Kostas et al., 2021; Sarkar & Etemad,

---

[*]Corresponding authors: {salarabb, ishapiro}@apple.com

2022; Mohsenvand et al., 2020; Gopal et al., 2021; Kiyasseh et al., 2021; Mehari & Strodthoff, 2022; Wu et al., 2020; Spathis et al., 2021; Tang et al., 2021; Yuan et al., 2023; Lai et al., 2023). While there have been recent efforts to apply SSL on wearable accelerometer data in free-living conditions (Spathis et al., 2021; Yuan et al., 2023), other SSL work have mostly used biosignal modalities such as PPG, ECG, and electroencephalogram (EEG) collected in clinical or controlled experimental settings (Cheng et al., 2020; Kostas et al., 2021; Sarkar & Etemad, 2022; Mohsenvand et al., 2020; Gopal et al., 2021; Kiyasseh et al., 2021; Mehari & Strodthoff, 2022; Lai et al., 2023). Developing foundation models for biosignals has several advantages as they: 1) require less amount of labeled data to reach to the same accuracy as supervised models, which significantly reduces the costs of experimental health studies (Cheng et al., 2020), 2) can be further fine-tuned on downstream targets, leading to better accuracy and requiring less computation resources given faster convergence compared to training from scratch (Hu et al., 2021), 3) provide signal-to-embedding models that can be used to calculate similarity scores between users or signals, and allow for faster information retrieval (Mitra & Craswell, 2018).

Here, using the large, longitudinal, multi-year Apple Heart and Movement Study (AHMS), we train biosignal foundation models with PPG and ECG. Our contributions are: 1) **Large-scale pre-training for biosignals:** To the best of our knowledge, this is the first work for pre-training foundation models on PPG and ECG biosignals using a large-scale dataset collected via wearable consumer devices with 141,207 participants. 2) **Self-supervised learning framework:** We combine techniques used in self-supervised learning for biosignals with techniques from other domains, such as computer vision. Our self-supervised pre-training has a stochastic participant level augmentation module, and the encoder is optimized with momentum training with a regularized InfoNCE loss. 3) **Studying information encoded by the pre-trained foundation models across various targets**: We show that pre-trained PPG and ECG embeddings contain information predictive of a broad range of demographic variables and health conditions. We studied the amount of information encoded separately in the PPG and ECG embeddings of pre-trained models with respect to various targets including demographics, health conditions and medications. 4) **Ablation studies**: We performed various ablation studies to evaluate the importance of participant level positive pair selection, to analyze distinctions in the pre-training and embeddings between PPG and ECG, to enable comparison with SimCLR (Chen et al., 2020) and BYOL (Grill et al., 2020) as examples of contrastive and non-contrastive pre-training frameworks on our dataset, to study the efficacy of different encoder architectures and model sizes, and to study the effect of different augmentation functions.

## 2 RELATED WORK

Self-supervised learning has recently gained popularity in various domains of deep learning. Our work is most related to a large body of self-supervised learning research using joint embedding architectures (JEA). JEAs are usually trained with a well-tailored loss to avoid representation collapse; contrastive losses such as InfoNCE avoid the collapse by contrasting various samples in the batch, and non-contrastive losses avoid the collapse using momentum training (Grill et al., 2020), or some form of regularization (Bardes et al., 2022), or both (Baevski et al., 2022). Self-supervised pre-trained models have been shown to encode significant amount of information about downstream targets without seeing any labels during pre-training (Chen et al., 2020; Baevski et al., 2020; Mohsenvand et al., 2020; Cheng et al., 2020; Kostas et al., 2021; Chen et al., 2021; Gopal et al., 2021; Kiyasseh et al., 2021; Mehari & Strodthoff, 2022; Baevski et al., 2022; Sarkar & Etemad, 2022; Oquab et al., 2023).

Self-supervised learning has also been proven to be effective in health applications, usually with small datasets of few hundred or few thousands participants and recorded in clinical or controlled experimental settings. These datasets include 12-lead ECG in hospital settings (Cheng et al., 2020; Gopal et al., 2021; Kiyasseh et al., 2021; Lan et al., 2021; Liu et al., 2021; Sarkar & Etemad, 2022; Mehari & Strodthoff, 2022; Diamant et al., 2022; Lai et al., 2023), EEG collected in controlled experimental setting (Mohsenvand et al., 2020; Cheng et al., 2020; Kostas et al., 2021; Banville et al., 2021) or PPG from few participants (Ghorbani et al., 2023). Recent work have used self-supervised learning for wearable accelerometer signals in free-living conditions and large dataset (Spathis et al., 2021; Yuan et al., 2023). However, to the best of our knowledge, our work presents the first study on pre-training foundation models from large-scale PPG and ECG data collected from wearable consumer devices.

## 3 SELF-SUPERVISED PRE-TRAINING

We use contrastive self-supervised learning to pre-train a deep neural network encoder, as described in Appendix Algorithm 1, and explain the components in details below.

### 3.1 POSITIVE PAIR SELECTION AND AUGMENTATIONS

Self-supervised learning with a JEA often requires positive pairs during pre-training. To motivate the model to extract information relevant to participant level physiology, we use a participant level positive pair selection strategy; the positive pairs are selected as augmented views of two different segments from the same participant. We denote segment $i$ for participant $s$ as $x_i^s$, therefore positive pairs are randomly selected pairs in form of $\{(x_i^s, x_j^s)|i \neq j\}$. The participant level positive pair selection significantly improves the quality and utility of the learned embeddings (Ablation 5.2.1).

Our augmentation module $T(\cdot)$ consists of a stochastic sequence of time-series augmentations, including crop, add Gaussian noise, time warp, magnitude warp, and channel swap, as explained in (Iwana & Uchida, 2021). We assign a configurable probability to each augmentation function in the augmentation set, and at each call of $T(\cdot)$, a sequence of augmentation functions in the augmentation set is applied given the random binary events drawn from the assigned probability values. In addition to this randomness, each augmentation has internal randomly selected hyperparameters, which we did not tune for the purposes of this study. Given various sources of randomness in the augmentation module, it covers a diverse range of distortions to the input segment, and we control the strength of distortions by changing the probability values. Our augmentation set and the assigned probability values for PPG are: {cut out: 0.4, magnitude warp: 0.25, add Gaussian noise: 0.25, channel permute: 0.25, time warp: 0.15}, and for ECG are: {cut out: 0.8, magnitude warp: 0.5, add Gaussian noise: 0.5, time warp: 0.3}. ECG segments have only one channel in our datasets (see 4.1), which is why we drop the channel permute augmentation for ECG. Also, to address the fact that ECG has less within-participant variability compared to PPG (Ablation 5.2.2), we assign higher probability values to our ECG augmentation module to introduce stronger signal distortions.

### 3.2 REGULARIZED INFONCE LOSS

We use InfoNCE (also known as NT-Xent) to maximize the mutual information between the positive pair representations while allowing for contrast to other positive pairs in each batch via the cross-entropy function to avoid representation collapse (Oord et al., 2019; Chen et al., 2020). In addition, to encourage a uniform span of features within the batch, we use Kozachenko-Leonenko (KoLeo) differential entropy estimator as regularization, as used in (Sablayrolles et al., 2019; Oquab et al., 2023). For each batch of embeddings $h$ from N positive pairs $(h_1, h_2)$, we define the model's objective as below:

$$L_{\text{contrastive}}^{(1,2)} = -\frac{1}{N} \sum_{i=1}^{N} \log \frac{\exp(sim(h_1^i, h_2^i)/\tau)}{\sum_{j=1}^{N} \mathbf{1}[j \neq i] \exp(sim(h_1^i, h_2^j)/\tau)}, \quad (1)$$

where $sim(\cdot, \cdot)$ is the cosine similarity function. KoLeo regularization is also calculated as:

$$L_{\text{KoLeo}}^{(1)} = -\frac{1}{N} \sum_{i=1}^{N} \log(min_{j \neq i} \left\| h_1^i - h_1^j \right\|^2). \quad (2)$$

The final objective is computed as the regularized InfoNCE loss:

$$L = \frac{1}{2}(L_{\text{contrastive}}^{(1,2)} + L_{\text{contrastive}}^{(2,1)}) + \frac{\lambda}{2}(L_{\text{KoLeo}}^{(1)} + L_{\text{KoLeo}}^{(2)}). \quad (3)$$

Both of the contrastive and regularization losses are calculated after $l2$-normalization of the embeddings and details can be found in Appendix Algorithm 1.

### 3.3 MOMENTUM TRAINING

While it is not necessary to use momentum training to avoid collapse given the contrastive component in our objective function, we use momentum training to create more mismatch between the positive

Table 1: Number of participants/segments, average number of calendar days per participant, and total dataset time span (time between the earliest to the latest recorded segment) in our pre-training datasets. We used the data from 80% of the participants for training, 10% for during training validation, and 10% for occasional post-training assessments.

|  | PPG | ECG |
| --- | --- | --- |
| Number of participants | 141,207 | 106,643 |
| Number of segments | 19,854,101 | 3,743,679 |
| Average number of calendar days per participant | 92.54 | 23.27 |
| Total dataset time span (days) | 890 | 1,240 |

pair representations, further encouraging the network to learn informative representations. This is particularly more helpful for signals such as ECG that have less within participant variability (Figure 2). We apply momentum training to both the encoder and projection head, where the online side of the JEA is updated via backpropagation and the momentum side is a lagging exponential moving average of the online side. The momentum update rule is $\xi \leftarrow \tau\xi + (1 - \tau)\theta$, where $\theta$ and $\xi$ denote the weights of the online side and the momentum side, respectively and $\tau$ is momentum update rate.

## 4 EXPERIMENTS

### 4.1 DATASETS

We used the PPG and ECG signals recorded on Apple Watch from participants in the Apple Heart and Movement Study (AHMS) (MacRae, 2021). AHMS is an ongoing research study designed to explore the links between physical activity and cardiovascular health, which is sponsored by Apple and conducted in partnership with American Heart Association and Brigham and Women's Hospital. To be eligible for the study, participants must be at least 18 years of age (21 in some locations), reside in the United States, have access to an Apple Watch, and provide informed consent electronically in the Apple Research app (Shapiro et al., 2023).

**PPG pre-training dataset:** Apple Watch intermittently and passively records green PPG signals during low-motion periods multiple times per day using light-emitting and light-sensitive diodes. The recorded PPG signals are 60 seconds in duration, sampled at 256Hz or 64Hz, and consist of four separate optical channels corresponding to different spatial combinations of transmitting and receiving diodes. We curated ∼20M PPG segments from ∼141K participants for our PPG dataset (Table 1). Random PPG segments were drawn from the full dataset given two conditions: 1) each participant had at least four segments in the pre-training dataset, 2) the number of segments per participant was as uniform as possible. PPG segments were pre-processed using dark subtraction (to reject signals introduced by ambient light), followed by bandpass filtering, down-sampling to 64Hz if needed and temporal channel-wise z-scoring for each segment.

**ECG pre-training dataset:** Apple Watch Series 4 or later enables users to record an ECG using a wrist-to-finger (Lead I) dry electrode sensor. One electrode is located on the crown and the other electrode is located on the back of the watch. When the user puts the watch on their wrist and makes contact using a finger of opposite hand, they complete a circuit and the ECG signal is recorded. The ECG recordings collected on Apple Watch are 30 seconds long at sampling frequency of 512Hz. We curated ∼3.75M ECG segments from ∼106K participants under the same two conditions for the PPG pre-training dataset. ECG signal recordings were pre-processed using Apple internal tools to produce 30-second strips equivalent to those stored locally in each participant's Health app, followed by down-sampling to 128Hz and temporal z-scoring for each segment. Brief statistics of our curated PPG and ECG datasets are in Table 1, AHMS demographics distribution is demonstrated in Appendix Fig. 5 and example PPG and ECG processed segments are shown in Appendix Fig. 6.

### 4.2 EVALUATION METRICS

**Linear probing for age, BMI and biological sex:** We perform linear probing for predicting self-reported age, body mass index (BMI), and biological sex (sex assigned at birth). For the classification tasks, we use ridge regression to predict scores for binarized targets (0/1) and we quantify the

Table 2: Downstream performance evaluation of PPG and ECG embeddings in predicting age, biological sex and BMI of participants, using two types of positive pair selection strategies during pre-training: 1) participant level, 2) segment level.

| Positive pair | Prediction task | PPG | | ECG | |
|---|---|---|---|---|---|
| | | AUC (pAUC) ↑ | MAE ↓ | AUC (pAUC) ↑ | MAE ↓ |
| Participant level (main) | Age classification | **0.976** (**0.907**) | - | **0.916** (**0.763**) | - |
| | Age regression | - | **3.19** | - | **6.33** |
| | BMI classification | **0.918** (**0.750**) | - | **0.797** (**0.612**) | - |
| | BMI regression | - | **2.54** | - | **3.72** |
| | Sex classification | **0.993** (**0.967**) | - | **0.951** (**0.841**) | - |
| Segment level | Age classification | 0.900 (0.762) | - | 0.783 (0.621) | - |
| | Age regression | - | 6.60 | - | 9.01 |
| | BMI classification | 0.766 (0.616) | - | 0.734 (0.557) | - |
| | BMI regression | - | 4.05 | - | 4.11 |
| | Sex classification | 0.870 (0.685) | - | 0.854 (0.683) | - |

performance with area under curve of receiver's operating curve (AUC), and partial AUC (pAUC) at false positive rate (FPR) of 10%. For regression tasks, we use ridge regression to predict continuous targets and we use mean absolute error (MAE) to quantify performance. For age, we classify ages above 50 versus below or equal to 50, as well as a regression estimate of the continuous age value. For BMI, we classify BMI values above 30 $kg/m^2$ versus BMI below or equal to 30 $kg/m^2$, as well as predict continuous BMI values. For biological sex, we classify male versus female. In all our analyses, we perform the linear probing at participant granularity: we mean-aggregate all the embeddings associated to each participant so that each participant contributes one and only one sample in the downstream training/evaluation. Also, the downstream training/evaluation splits are stratified based on participants such that the evaluation split's participants have no overlap with those in the training split.

**Linear probing for targets from AHMS survey questions:** During AHMS, participants fill out multiple questionnaires containing various questions regarding their historical health record and demographics. The response to these questions are usually in form of 'yes' or 'no' for whether the participant has had a history of a health condition (e.g., asthma), or whether they take specific medications (e.g., anti-depressants), or regarding their lifestyle habits (e.g., smoking). Detailed description of questions and how we define targets is in Appendix A.2. To have a baseline, we compare the predictive ability of the pre-trained encoder embeddings vs. baseline features that include age, biological sex, BMI, and ethnicity, in addition to average heart rate and standard deviation heart rate that are derived from PPG and are known to be well informative of certain conditions. Similar to above, we use ridge regression to predict scores for binarized targets (0/1) where training/evaluation splits are stratified based on participants. We quantify linear classification of these targets using AUC from PPG embeddings, and compare these with classification AUC using a baseline feature set consisting only of self-reported demographic information (age, BMI, biological sex and race/ethnicity) and measured heart rate.

**Smooth effective rank:** Smooth effective rank is an unsupervised metric quantifying the entropy of the singular value distribution of the embeddings in a given batch (Roy & Vetterli, 2007; Garrido et al., 2023), and can be used as a crude proxy for downstream evaluations without requiring any labels, as it has been shown to positively correlate with downstream evaluations (Garrido et al., 2023). We calculated smooth effective rank as its average for the batches of size 256 in the pre-training validation split.

## 4.3 IMPLEMENTATION DETAILS

Our default encoder is an EfficientNet-style 1D convolutional neural network (Ablation 5.2.4) with 16 mobile-inverted bottleneck blocks with squeeze-and-excitation (Tan & Le, 2020) and we used 256-dimensional embedding (the representation vector after the deep encoder) for all models used in this study. The encoder had 3.3M parameters for PPG and 2.5M for ECG. The projection head was a multi-layer perceptron with one hidden layer of 1024 units, taking the 256-dimensional embedding

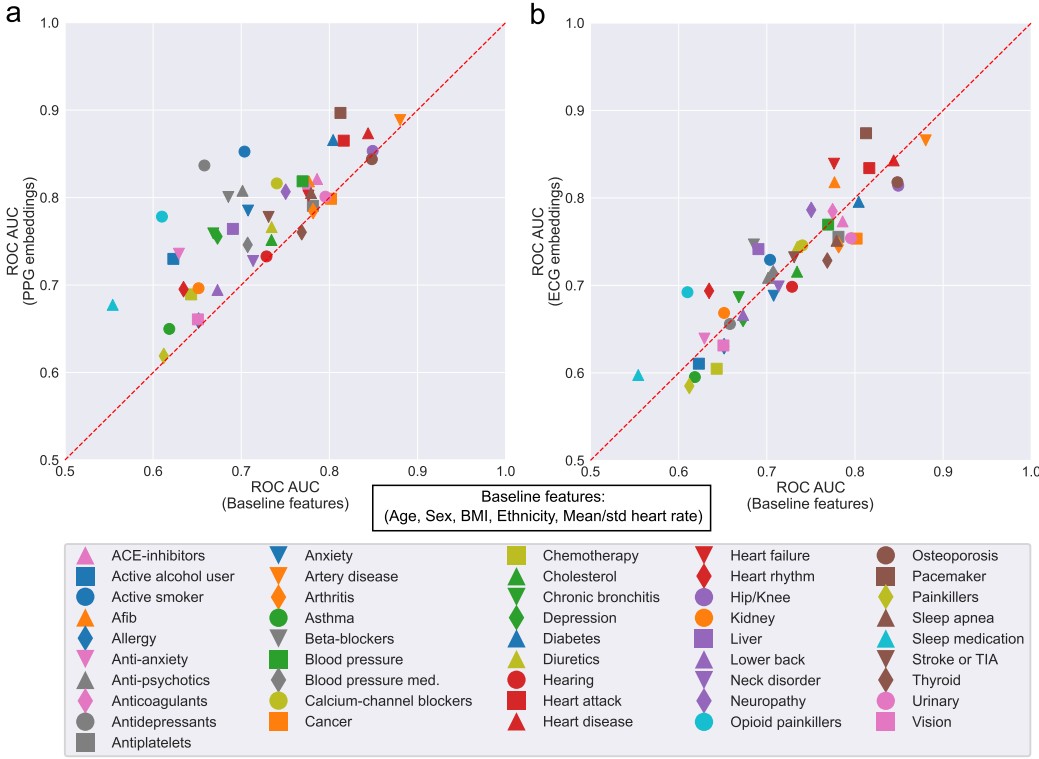

Figure 1: PPG and ECG foundation models encode participants' health information. The comparison of linear probing evaluation for targets from AHMS survey questions (**a**) using PPG embeddings, (**b**) and using ECG embeddings, versus baseline features is shown. Each marker represents one of the targets from AHMS questionnaire, the y-axis represents the ROC AUC of binary classification using the embeddings, and the x-axis represents that for the baseline features. The marker color and shapes are selected randomly, and are described in the legend.

to a 128-dimensional representation subspace where the loss is calculated in. For InfoNCE, we used temperature value of 0.04 for both PPG and ECG modalities, and the weight for the KoLeo regularization in our objective function was set to 0.1. The models were trained using Adam optimizer with gradient descent updates distributed across 32 A100 GPUs. Other implementation details is in Appendix A.1.

## 5  RESULTS

### 5.1  PPG AND ECG FOUNDATION MODELS ENCODE PARTICIPANTS HEALTH INFORMATION

We analyzed the amount of information encoded in the embeddings (the representation after the deep encoder) via linear probing. First, we evaluated linear classification and regression performance for predicting age, biological sex and BMI for both PPG and ECG embeddings, summarized in Table 2 (see 4.2 for details). We observed that PPG embeddings were more predictive of these targets, suggesting that periodically-collected background PPG signals may contain more information predictive of downstream health predictions compared to user-initiated ECG signals . Second, we evaluated linear classification performance for many binary targets derived from AHMS self-reported questionnaires using PPG and ECG embeddings, and compared these with the corresponding classifier performance achieved using a baseline feature set consisting only of self-reported demographic information (age, BMI, biological sex and race/ethnicity) and measured heart rate (see 4.2 for details). Fig. 1a and Fig. 1b show the ROC AUC for predicting these targets using PPG and ECG embeddings compared to the baseline features (also see Appendix Table 9). We made multiple observations: 1) PPG embeddings almost always performed better than baseline features in predicting

downstream conditions, indicating that the PPG foundation model embeddings readily encode information regarding participant health conditions, and this information is beyond what that can be predicted from the participant demographics and heart rate. This gives us confidence that the encoder is not simply using heart rate and demographics (which can be inferred from the biosignals with surprising accuracy as shown in Table 2) as a 'shortcut' to predict whether the participant has an age-related condition such as heart disease. 2) While there are various health conditions that are better predicted with ECG compared to the baseline, it appears that ECG embeddings encode less information regarding participant health compared to PPG embeddings. This could have different reasons: ECG signals may contain less information specific to these conditions, and/or our pre-training framework is more optimal for PPG compared to ECG (see Ablation 5.2.2 and Discussion). The gap in performance is not explained by the different number of segments in the PPG and ECG datasets (20m vs. 3.75m), as we observed that a model trained on a smaller PPG dataset which has similar number of segments to the ECG dataset still had significantly better performance in predicting demographics and health conditions (Appendix Table 10).

## 5.2 ABLATION STUDIES

### 5.2.1 POSITIVE PAIR SELECTION

We did an ablation analysis regarding the importance of participant level positive pair selection by changing this to segment (instance) level during the pre-training phase. Segment level positive pair selection is conventionally used for self-supervised learning (Chen et al., 2020; Baevski et al., 2020; Mohsenvand et al., 2020; Cheng et al., 2020; Kostas et al., 2021; Chen et al., 2021; Gopal et al., 2021; Kiyasseh et al., 2021; Mehari & Strodthoff, 2022; Baevski et al., 2022; Sarkar & Etemad, 2022; Oquab et al., 2023), where a positive pair is two augmented views of an individual segment/sample. We observed that consistently for both PPG and ECG, pre-trained models with participant level positive pairs were significantly more predictive of downstream targets (Table 2). While the evaluations in Table 2 are at participant granularity (see 4.2), we observed similar improvements for evaluations at segment granularity (Appendix Table 11).

### 5.2.2 DISTINCTIONS IN PPG AND ECG PRE-TRAINING AND EMBEDDINGS

We observed meaningful differences in self-supervised pre-training for PPG compared with ECG signals. To better understand the distinctions, we studied the PPG vs. ECG pre-training metrics and the resulting embeddings:

**Visual inspection of t-sne representations**: We performed t-sne dimensionality reduction (Maaten & Hinton, 2008) on 200 segment embeddings, consisting of 20 segments randomly drawn from 10 random participants, for both PPG and ECG modalities. First, as expected, we observed that participants form clusters in the t-sne subspace (Fig. 2a). However, visually inspecting these plots (only one shown here), we observed that ECG participant clusters tended to be denser than those for PPG. This qualitatively shows that ECG segments, and thus ECG embeddings, are more similar within a particular participant as opposed to that for PPG. This is one of the reasons our augmentation cascade (see 3.1) has higher probability values for each augmentation for ECG compared to PPG, to make the pre-training more difficult for ECG by introducing a stronger distortion to the original segments.

**InfoNCE loss:** In line with the previous observation, we looked at the validation InfoNCE loss for ECG vs. PPG, and observed that the InfoNCE loss was significantly smaller for ECG compared with PPG (Fig. 2b). This further corroborates that pre-training using InfoNCE is an easier task for ECG as opposed to PPG, perhaps due to the fact that it is easier to distinguish participants from their ECG segments versus their PPG segments.

**Dispersion ratio:** To quantify what we qualitatively observed from visually inspecting reduced t-sne dimensions, we used a dispersion ratio metric that is computed as the ratio between within participant variability vs. across participant variability. Fig. 2c represents the dispersion ratio for each of the 256 features in the embeddings for PPG vs. ECG. We observed that, in line with previous observations, ECG embeddings had statistically smaller dispersion ratio across the population compared to that for PPG embeddings.

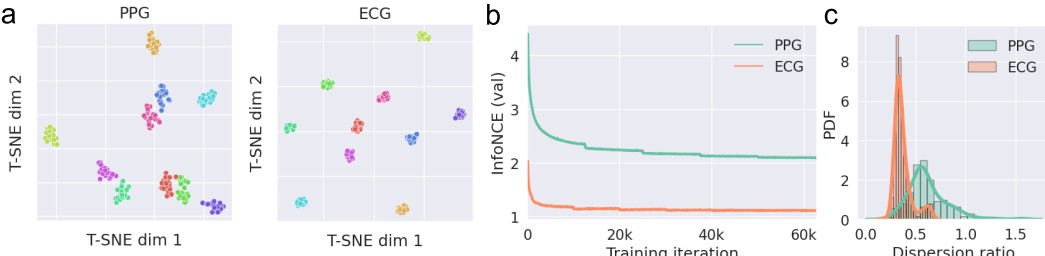

Figure 2: Distinctions in PPG and ECG pre-training and embeddings. **a.** T-SNE representations for 20 random embeddings drawn from 10 random participants for PPG and ECG. **b.** InfoNCE validation loss for PPG vs. ECG, where training iteration represents a global gradient descent update across all GPUs. **c.** Dispersion ratio probability density function (PDF) calculated for each feature of the 256-dimensional PPG and ECG embeddings across the population. Dispersion ratio quantifies within participant variability to across participant variability.

These results together demonstrate that ECG and PPG signals have inherent distinctions with respect to the participant- and health-specific information contained in their embeddings. While our pre-training generalizes well to both modalities, follow up methods could ideally use modality-specific frameworks to better take into account such distinctions.

### 5.2.3 PRE-TRAINING FRAMEWORK

We compared our pre-training framework with our variation of SimCLR and BYOL, as popular frameworks for contrastive and non-contrastive self-supervised learning. In fact, to make the comparison fair, almost all other training details were similar between these methods, including participant level positive pair selection even though the original frameworks had segment level positive pair selection (Chen et al., 2020; Grill et al., 2020), augmentation module, or other important implementation details in 4.3. For comparisons, we used smooth effective rank as a crude proxy of general downstream performance, and downstream evaluations for specific targets of age, biological sex and BMI. In general, we observed that contrastive frameworks were more robust and resulted in more informative embeddings (Table 3 and Appendix Table 12). Also, we saw that our pre-training framework was better than these two baseline variations across most evaluations. We observed that for both PPG and ECG modalities, removing KoLeo regularization from our objective function resulted in reduced evaluation metrics (Table 3 and Appendix Table 12), demonstrating its impact on the model's performance. We also want to emphasize that, as discussed in 5.2.2, unsupervised evaluation metrics such as InfoNCE and smooth effective rank are not directly comparable between signal modalities. In fact, we believe low InfoNCE and high smooth effective rank are necessary for good downstream performance but not sufficient – these unsupervised objective/metrics are correlated with downstream performance but the mapping/relationship is not necessarily monotonic. For instance, we observed that ECG embeddings had lower InfoNCE (Figure 2) and higher smooth effective rank (Table 3) compared to PPG embeddings, while PPG was more predictive of demographics and health conditions (Figure 1 and Table 2).

### 5.2.4 ENCODER ARCHITECTURE

We used different encoder architectures within our pre-training framework to study their importance in extracting informative embeddings. We compared our default 1D-EfficientNet with our variation of 1D-ResNet and 1D-ViT (see A.1). For comparisons, we used smooth effective rank as a crude proxy of general downstream performance, and observed that different encoder architectures, with different model sizes, can achieve relatively similar performance, demonstrating that the performance is not unique to the 1D-EfficientNet encoder architecture (Table 4). We observed that 1D-EfficientNet model yielded similar performance to these alternative encoders with significantly smaller number of parameters, which was one of the main reasons we picked 1D-EfficientNet as our default backbone encoder given less memory footprint when potentially running on a wearable device. An interesting future direction is scaling up the model size further, particularly with transformer-based models (e.g., 1D-ViT) given their scalability (Kaplan et al., 2020), and larger datasets to study its effect on downstream performance.

Table 3: Smooth effective rank calculated for different pre-training frameworks (higher is better).

| Pre-training | PPG | ECG |
|---|---|---|
| Ours | **104.13** | **113.82** |
| Ours (no KoLeo) | 101.29 | 108.84 |
| SimCLR (our variation) | 98.87 | 103.65 |
| BYOL (our variation) | 51.18 | 63.67 |

Table 4: Smooth effective rank (SER) and number of model parameters (Size) calculated for different encoder architectures within our pre-training framework.

| Encoder | PPG | | ECG | |
|---|---|---|---|---|
| | SER ↑ | Size ↓ | SER ↑ | Size ↓ |
| 1D-EfficientNet (our variation) | 104.13 | **3.3M** | 113.82 | **2.5M** |
| 1D-ResNet (our variation) | 95.81 | 16.9M | 111.92 | 16.9M |
| 1D-ViT (our variation) | **104.89** | 7.2M | **114.62** | 7.2M |

### 5.2.5 AUGMENTATIONS

While we did not comprehensively tune our augmentation module (see Appendix A.1), we studied the effect of single augmentation functions for each modality. To do so, for both PPG and ECG, we kept each of the augmentation functions in isolation separately (with probability 1), and repeated our pre-training framework from scratch. We observed that PPG pre-training was more sensitive to the choice of single augmentation functions (more variance in performance), and channel permute and cut out were the most effective augmentation functions in isolation for PPG and ECG, respectively (Appendix Fig. 7). Future work can study more optimal ways to design modality-specific augmentation modules.

## 6 DISCUSSION, LIMITATIONS AND FUTURE WORK

Pre-training foundation models using unlabeled data has been proven successful in various domains of deep learning such as natural language processing and computer vision. In this work, we have pre-trained foundation models for PPG and ECG, as examples of common biosignals from wearable devices, using self-supervised learning and a large longitudinal dataset. We have shown that our pre-trained foundation models readily encode participant demographics with high accuracy (see A.4 for further discussion), as well as encode information predictive of a broad range of self-reported health conditions and medication categories. It is worth noting that our target labels are opportunistically constructed from participants' self-reported survey. Therefore, the labels used for training and evaluation the downstream classifiers are sub-optimal and future work can quantify these predictions using more accurate labels. Also, we would like to note that while PPG/ECG embeddings are predictive of health conditions, other biomarkers including but not limited to heart rate, heart rate variability (HRV), and physical activity still provide valuable insight regarding one's health status, and their relationship to PPG and ECG embeddings should be examined further in future work. Our pre-training framework generalizes to PPG and ECG biosignals, however, we believe future work could gain improvements by incorporating modality-specific design choices. Future work can also investigate the longitudinal changes in PPG and ECG embeddings, and whether accounting for those can improve downstream predictions. Another future area of investigation is multi-modal pre-training by: 1) using a multi-modal encoder that takes multiple modalities (e.g., PPG and ECG) as input, 2) or CLIP-style multi-modal pre-training by supervising one modality with another one (Radford et al., 2021), or training multiple encoders for multiple modalities simultaneously (Girdhar et al., 2023), using a contrastive loss. Last but not least, we observed that the choice of positive pairs significantly affects the quality of the embeddings; future work can investigate the efficacy of different positive pair selection strategies on different health-related targets, by accounting for temporal and other physiological information.

ACKNOWLEDGMENTS

We would like to thank participants in Apple Heart and Movement Study, Calum MacRae, MD, PhD, and study staff at The Brigham and Women's Hospital, a Harvard affiliate, without whom this work would not have been possible. We also thank Jen Block, Joe Futoma, Sarvesh Kirthivasan, Andy Long and Chris Sandino for providing valuable feedback regarding the manuscript, Zach Ellison and Mayank Gupta for the training platform support, Lindsay Hislop, Eduardo Martinez Montes and Laura Rhodes for publication coordination.

REPRODUCIBILITY STATEMENT

The aggregated data that support the findings of this study can be made available on request from the corresponding author. Request for data will be evaluated and responded to in a manner consistent with the specific language in the study protocol and informed consent form (ICF). Based on the language within the IRB approved ICF for the Apple Heart and Movement Study, we are unable to share sensor data collected in the study. Similarly, code for all data analyses may be available upon request from the corresponding author. Requests for code will be evaluated and responded to in a manner consistent with policies intended to protect participant confidentiality and language in the study protocol and ICF.

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

# A  APPENDIX

## A.1  SUPPLEMENTARY TRAINING DETAILS

**Pre-training framework, encoder architectures and other implementation details**: Pseudo-code of our pre-training framework is in Algorithm 1 and it is visually shown in Fig. 3. Also, brief description of our encoder architecture is shown in Fig. 4. For BYOL, the prediction head was a multi-layer perceptron with one hidden layer of 1024 units. We observed that batch normalization was necessary to prevent collapse for BYOL, and for consistency, all our models had batch normalization in their projection and prediction (if any) heads. Throughout this study, we used constant momentum update rate of 0.99 for our pre-training framework and BYOL. For all model training in this study (unless otherwise stated), we used a batch size of 256, and the initial learning rate was 0.001 (0.00025 for BYOL), and we used step learning rate scheduling for a faster convergence. Regarding batch size in our pre-training, while early contrastive self-supervised learning works have shown that larger batch size is necessary to improve performance (Chen et al., 2020; 2021), follow up works removed this dependency by changing the pre-training (Grill et al., 2020; Zbontar et al., 2021), and a recent work has shown that it is possible to train SimCLR on ImageNet with smaller batch sizes without an important drop in performance (Bordes et al., 2023). We also did not see meaningful change in performance with larger batch sizes in our early experiments. We did not comprehensively tune our augmentation module probabilities, we started with a preliminary set of probabilities in our early experiments based on prior work (Cheng et al., 2020; Tang et al., 2021), for instance, cut out was shown to be the most effective augmentation in (Cheng et al., 2020) and we assigned a higher probability to it. We did not tune these probabilities afterwards and the only change made was to make the ECG augmentation probabilities stronger (2x probability) to allow for more mismatch between positive pair representations. For the 1D-ResNet and 1D-ViT encoder architectures in 5.2.4, we almost used similar training details as explained for 1D-EfficientNet in 4.3, with the only difference that linear learning rate warm up from $50\%$ of max learning rate for the first 10 epochs was employed for training the 1D-ViT encoder to ensure a more stable and optimal training (Xiong et al., 2020). Our 1D-ResNet was adapted from ResNet-34 with residual connections (He et al., 2015) for 1D time-series, and our 1D-ViT had a 6-layer convolutional neural network for token embedding (receptive field around 178 input samples), resulting in 60 tokens with 256 dimensions, followed by a 8-layer transformer with 8 attention heads and 1024 MLP feedforward dimension, and finally global average pooling on output tokens to get the final 256-D embedding. In our experiments, we observed that convolutional neural network token embedding, as used in prior models for audio signals (Baevski et al., 2020; 2022) outperformed linear token embeddings used in the original ViT work (Dosovitskiy et al., 2021).

## A.2  SUPPLEMENTARY DATASET DETAILS

**AHMS demographics distribution**: The distribution of demographic variables in AHMS is shown for the participants that opted to answer the survey questions regarding each of age, BMI (derived from weight and height), biological sex, and ethnicity in Fig. 5.

**Representative PPG/ECG signals**: Representative examples of processed PPG and ECG signals are shown in Fig.6.

**AHMS Survey**: AHMS survey is formed of multiple questionnaires which participants fill out one or multiple times over the course of their participation in the study; for all the analyses in this study, we used participants' latest entry before 01/28/2022. Tables 5 and 6 contain AHMS survey questions about medical conditions and medications, respectively, in addition to the corresponding target labels used in Fig. 1. Table 7 includes AHMS survey questions about drinking and smoking habits, and Table 8 defines our logic to summarize these questions into binary labels for the related targets used in Fig. 1.

## A.3  SUPPLEMENTARY RESULTS

**Linear probing evaluation of targets from AHMS survey questions**: Table 9 includes the linear probing evaluation of targets from AHMS survey questions shown in Fig. 1, using PPG embeddings, ECG embeddings and baseline features.

**Comparison of downstream performance for different pre-training frameworks**: Table 12 includes the downstream evaluation of age, biological sex and BMI for our pre-training framework and our variation of SimCLR and BYOL.

**Effect of single augmentation functions in pre-training**: Figure 7 represents the effect of single augmentation functions in pre-training for both PPG and ECG modalities.

## A.4 SUPPLEMENTARY DISCUSSION

**Predicting demographic variables via representation learning from biosignals in prior work**: Prior work have looked into predicting certain demographic variables via representation learning from biosignals: 1) Using activity, sleep, and heart rate, BMI was predicted with ROC AUC of 0.697 (with the binary class cut-off at 30 kg/m$^2$ similar to us), which is significantly smaller than that with our PPG/ECG embeddings in Table 2. Also, age was predicted with ROC AUC of 0.701 (with the binary class cut-off at 31 years) which is not directly comparable to our evaluation due to different cut-offs (Hallgrímsson et al., 2018). 2) A prior work pre-trained models using wearable accelerometer and heart rate data in free-living conditions (Spathis et al., 2021), and predicted age, BMI, and biological sex with ROC AUC of 0.676, 0.694, and 0.934, respectively, where the binary class cut-off was set to the median for continuous variables such as age and BMI. While the biological sex prediction is more accurately predicted with PPG and ECG embeddings in Table 2, the age and BMI predictions are not directly comparable due to different factors including different cut-off for binary labels and different dataset distribution, e.g., the distribution of the age in AHMS (Fig. 5) is different from the dataset used in this prior work (age ranging from 35 to 65). 3) Using the heart rate time-series data, biological sex was predicted with 0.703 and 0.672 Micro and Macro F1 score (Wu et al., 2020) - these reported numbers are lower than that for PPG embeddings (0.973 and 0.969, respectively) and for ECG embeddings (0.892 and 0.869, respectively). This prior work also reports age prediction which is not directly comparable to us due to different class buckets.

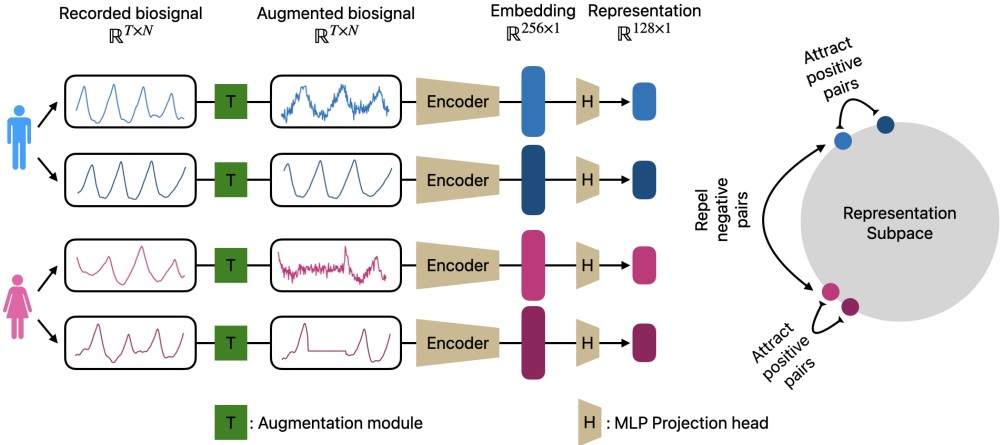

Figure 3: High-level visualization of our pre-training framework shown for a mini-batch containing 2 participants and 4 segments. Augmented views of recorded biosignals are passed through an encoder, followed by an MLP projection head to get the representations. The representations are used to calculate the contrastive loss which attracts the positive pairs while repelling the negative pairs. Positive pairs are formed as different segments of the same participant. The momentum updates and KoLeo regularization are not shown on the figure.

---

**Algorithm 1:** Pseudocode of our pre-training framework in PyTorch-like style.

```
# f, f_m:  online and momentum encoder networks, including the MLP
  projection heads
# m:  momentum rate
# t:  temperature
# w:  lambda weight for KoLeo regularization
# loss_cont:  contrastive loss
# loss_reg:  KoLeo regularization penalty
# initialize momentum encoder with online encoder weights
f_m.params = f.params
# training loop
for x1, x2 in loader:  # load a batch of positive pairs from N
  participants
   # get the augmented views of the segments
   x1, x2 = aug(x1), aug(x2)
   # get the representations from the online and momentum encoders
   h1, h2 = f(x1), f(x2)
   h1_m, h2_m = f_m(x1), f_m(x2)
   # stop gradients for momentum encoder
   h1_m, h2_m = h1_m.detach(), h2_m.detach()
   # calculate the regularized InfoNCE loss (after l2-normalization of
    embeddings inside loss_cont and loss_reg)
   loss = loss_cont(h1, h2_m, t) + loss_cont(h2, h1_m, t)
        + w * (loss_reg(h1) + loss_reg(h2))
   # backprop through the online encoder
   optimizer.zero_grad()
   loss.backward()
   optimizer.step()
   # momentum update for the momentum encoder
   f_m.params.data = m * f_m.params.data + (1 - m) * f.params.data
```

---

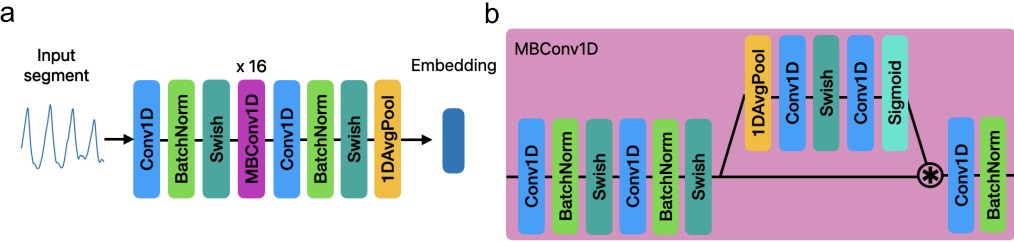

Figure 4: Our EfficientNet-style encoder architecture, adapted from (Tan & Le, 2020) for time-series input. **a.** Encoder architecture with convolutional blocks shown as Conv1D, batch normalization as BatchNorm, Swish activation as Swish, mobile inverted bottleneck block as MBConv1D, average pooling as 1DAvgPool. **b.** The internal architecture of MBConv1D, where Sigmoid activation is shown as Sigmoid and asterisk represents element wise multiplication.

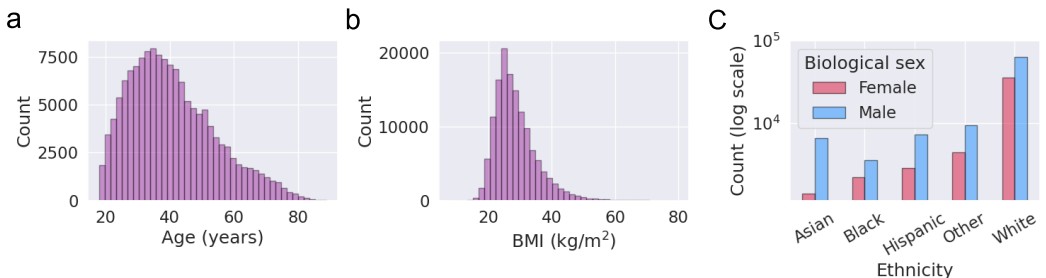

Figure 5: The distribution of demographic variables in AHMS shown for self-reported **a.** age, **b.** BMI, **c.** biological sex and ethnicity.

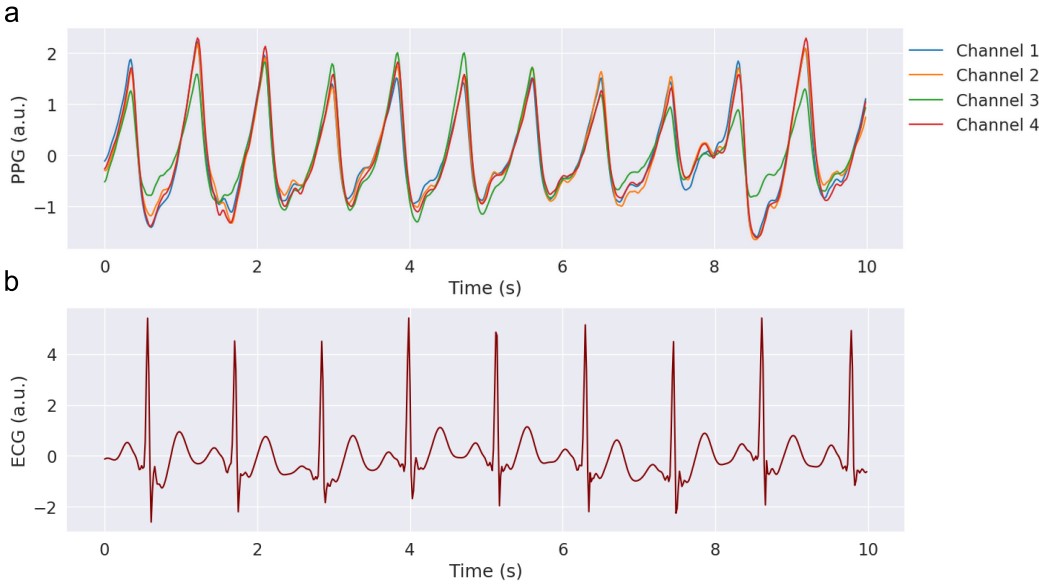

Figure 6: A representative example for the processed segments shown for **a.** PPG, **b.** ECG, in arbitrary units (a.u.) for 10 seconds.

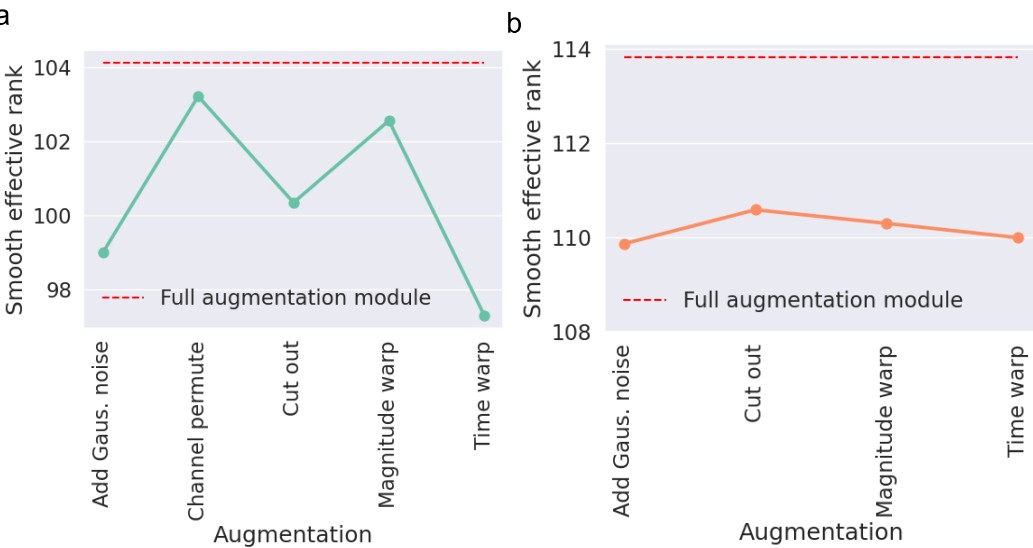

Figure 7: The effect of single augmentation functions in pre-training for **a.** PPG, **b.** ECG. For each augmentation function, we kept it in isolation and probability 1 in our augmentation module, and repeated pre-training. We then calculated and compared smooth effective rank with other augmentation functions and the full augmentation module.

Table 5: AHMS survey questions about medical conditions. The main question is in form of 'Have you ever been diagnosed with any of the following conditions?' and participants can answer 'Yes' or 'No' or 'I prefer not to answer' or 'I don't know'. The question for vision and hearing loss is different, which we explicitly mention in the corresponding rows. Third column indicates the number of left out participants for evaluation – the reason for variations is that for each target we exclude participants whose answers were 'I prefer not to answer' or 'I don't know' or missing.

| Target label | Medical condition | N (test) |
|---|---|---|
| Heart attack | Heart attack (myocardial infarction) | 16,832 |
| Heart disease | Coronary heart disease or angina pectoris | 16,715 |
| Blood pressure | High blood pressure (hypertension) | 16,574 |
| Stroke or TIA | Stroke (cerebral hemorrhage, cerebral thrombosis) or transient ischemic attack (ministroke) | 16,850 |
| Afib | Atrial fibrillation | 16,585 |
| Heart rhythm | Heart rhythm problem other than atrial fibrillation | 16,428 |
| Pacemaker | Pacemaker | 16,902 |
| Artery disease | Peripheral artery disease | 16,613 |
| Heart failure | Heart failure | 16,851 |
| Diabetes | Diabetes | 16,770 |
| Cholesterol | High cholesterol | 16,499 |
| Arthritis | Arthritis | 16,598 |
| Hip/Knee | Hip or knee replacement | 16,932 |
| Lower back | Low back disorder or other chronic back defect | 16,682 |
| Neck disorder | Neck disorder or other chronic neck defect | 16,739 |
| Osteoporosis | Osteoporosis | 16,722 |
| Asthma | Asthma | 16,812 |
| Chronic bronchitis | Chronic bronchitis, chronic obstructive pulmonary disease, or emphysema | 16,818 |
| Allergy | Rhinitis, hay fever, eye inflammation, dermatitis, food allergy or other allergy (allergic asthma excluded) | 16,729 |
| Kidney | Kidney problems | 16,789 |
| Thyroid | Thyroid disease | 16,706 |
| Cancer | Cancer | 16,836 |
| Liver | Cirrhosis of the liver | 16,848 |
| Urinary | Urinary incontinence | 16,804 |
| Neuropathy | Neuropathy | 16,589 |
| Depression | Depression | 16,457 |
| Anxiety | Anxiety disorder | 16,380 |
| Hearing | Do you have hearing loss? | 15,476 |
| Vision | Do you have vision loss? | 16,295 |

Table 6: AHMS survey questions about medications. The main question is in form of 'Do you currently take any of the following types of medications?' and participants can answer 'Yes' or 'No' or 'I prefer not to answer'. The formatting for the medications is similar to their presentation in the tudy, but may not exactly match the format in the study application. Third column indicates the number of left out participants for evaluation – the reason for variations is that for each target we exclude participants whose answers were 'I prefer not to answer' or missing. Third party trademarks used herein are trademarks of their respective owners.

| Target label | Medications | N (test) |
|---|---|---|
| ACE-inhibitors | ACE-inhibitors or ARBs (for blood pressure) such as captopril, enalapril, lisinopril, losartan, ramipril, or valsartan | 11,043 |
| Anti-anxiety | Anti-anxiety aids such as alprazolam (Xanax®), clonazepam (Klonopin®), clorazepate (Tranxene®), diazepam (Valium®), or lorazepam (Ativan®) | 11,047 |
| Anti-psychotics | Anti-psychotics such as haloperidol (Haldol®), aripiprazole (Abilify®), risperidone (Risperdal®), quetiapine (Seroquel®), olanzapine (Zyprexa®), clozapine (Clozaril®), or lurasidone (Latuda®) | 11,076 |
| Anticoagulants | Anticoagulants (blood thinners) such as warfarin (Coumadin®), apixaban (Eliquis®), betrixaban (Bevyxxa®), dabigatran (Pradaxa®), edoxaban (Lixiana®), or rivaroxaban (Xarelto®) | 11,075 |
| Antidepressants | Antidepressants such as amitriptyline (Elavil®), bupropion (Wellbutrin®), citalopram (Celexa®), duloxetine (Cymbalta®), escitalopram (Lexapro®), fluoxetine (Prozac®), paroxetine (Paxil®), mirtazapine (Remeron®), sertraline (Zoloft®), or venlafaxine (Effexor®) | 11,073 |
| Antiplatelets | Antiplatelets (blood thinners) such as aspirin, clopidogrel (Plavix®), prasugrel (Effient®), or ticagrelor (Brilinta®) | 11,073 |
| Beta-blockers | Beta-blockers (for blood pressure or heart rhythm) such as atenolol (Tenormin®), bisoprolol (Zebeta®), carvedilol (Coreg®), labetalol, metoprolol (Lopressor®, Toprol-XL®), nadolol (Corgard®), nebivolol (Bystolic®), propranolol (Inderal®), or sotalol (Betapace®) | 11,039 |
| Blood pressure med. | Other medications for lowering blood pressure such as clonidine, hydralazine, minoxidil, or sacubitril/valsartan (Entresto®) | 11,010 |
| Calcium-channel blockers | Calcium-channel blockers (for blood pressure or heart rhythm) such as amlodipine (Norvasc®), diltiazem, or verapamil | 11,019 |
| Chemotherapy | Certain types of chemotherapy such as carboplatin, cisplatin, oxaliplatin, vincristine, or vinblastine | 11,084 |
| Diuretics | Diuretics (water pills) such as chlorthalidone, furosemide (Lasix®), hydrochlorothiazide, or spironolactone | 11,067 |
| Opioid painkillers | Opioid painkillers such as codeine, fentanyl, hydrocodone, hydromorphone (Dilaudid®), meperidine (Demerol®), morphine, oxycodone, Percocet®, or Vicodin® | 11,090 |
| Painkillers | Non-steroidal anti-inflammatories (painkillers) such as aspirin, celecoxib (Celebrex®), diclofenac (Cambia®), ibuprofen (Motrin®/Advil®), or naproxen (Aleve®) | 11,069 |
| Sleep medication. | Sleeping aids such as eszopiclone (Lunesta®), zaleplon (Sonata®), or zolpidem (Ambien®) | 11,057 |

Table 7: AHMS survey questions about drinking and smoking habits. These are standardized questions from the AUDIT-C questionnaire (Bush et al., 1998) and All of US research program (Denny et al., 2019; Ramirez et al., 2022).

| # | Question | Answer choices |
|---|----------|----------------|
| Q1 | In your entire life, have you had at least 1 drink of any kind of alcohol, not counting small tastes or sips? | 'Yes'/'No'/'I don't know'/'I prefer not to answer' |
| Q1b | [If yes to Q1] How often did you have a drink containing alcohol in the past year? | 'Never'/'Monthly or less'/'Two to four time a month'/'Two to three times a week'/'Four or more times a week'/'I prefer not to answer' |
| Q1c | [If yes to Q1] On a typical day when you drink, how many drinks do you have? | '1 or 2'/'3 or 4'/'5 or 6'/'7 to 9'/'10 or more'/'I prefer not to answer' |
| Q2 | Have you smoked at least 100 cigarettes in your entire life? | 'Yes'/'No'/'I don't know'/'I prefer not to answer' |
| Q2b | [If Yes or Do not know to Q2] Do you now smoke cigarettes every day, some days, or not at all? | 'Every day'/'Some days'/'Not at all'/'I prefer not to answer' |

Table 8: Our logic for defining drinking and smoking targets from questions in Table 7. Third column indicates the number of left out participants for evaluation – the reason for variations is that for each target we exclude participants whose answers did not conclude in our binary 'yes' or 'no' mappings or were missing.

| Target label | Logic | N (test) |
|--------------|-------|----------|
| Active alcohol user | 'Yes': answer to Q1 is 'Yes' and answer to Q1b is 'Two to three times a week'/'Four or more times a week'
'No': answer to Q1 is 'No', or answer to Q1b is 'Never'/'Monthly or less'/'Two to four time a month' | 16, 236 |
| Active smoker | 'Yes': answer to Q2 is 'Yes' and answer to Q2b is 'Every day'/'Some days'
'No': answer to Q2 is 'No', and answer to Q2b is 'Not at all' | 16,214 |

Table 9: Linear probing evaluation of targets shown in Fig 1.

| Name | ROC AUC (PPG embeddings) | ROC AUC (ECG embeddings) | ROC AUC (Baseline features) |
|---|---|---|---|
| ACE-inhibitors | 0.821 | 0.773 | 0.786 |
| Active alcohol user | 0.730 | 0.611 | 0.623 |
| Active smoker | 0.853 | 0.729 | 0.704 |
| Afib | 0.818 | 0.818 | 0.777 |
| Allergy | 0.659 | 0.631 | 0.652 |
| Anti-anxiety | 0.736 | 0.639 | 0.629 |
| Anti-psychotics | 0.808 | 0.709 | 0.701 |
| Anticoagulants | 0.811 | 0.785 | 0.775 |
| Antidepressants | 0.837 | 0.656 | 0.658 |
| Antiplatelets | 0.791 | 0.756 | 0.781 |
| Anxiety | 0.785 | 0.688 | 0.708 |
| Artery disease | 0.889 | 0.866 | 0.880 |
| Arthritis | 0.785 | 0.746 | 0.782 |
| Asthma | 0.650 | 0.595 | 0.618 |
| Beta-blockers | 0.801 | 0.747 | 0.685 |
| Blood pressure | 0.819 | 0.769 | 0.770 |
| Blood pressure med. | 0.746 | 0.714 | 0.707 |
| Calcium-channel blockers | 0.816 | 0.746 | 0.740 |
| Cancer | 0.799 | 0.754 | 0.802 |
| Chemotherapy | 0.689 | 0.605 | 0.643 |
| Cholesterol | 0.751 | 0.716 | 0.734 |
| Chronic bronchitis | 0.759 | 0.687 | 0.668 |
| Depression | 0.756 | 0.662 | 0.673 |
| Diabetes | 0.866 | 0.796 | 0.804 |
| Diuretics | 0.766 | 0.744 | 0.734 |
| Hearing | 0.733 | 0.698 | 0.729 |
| Heart attack | 0.865 | 0.834 | 0.816 |
| Heart disease | 0.874 | 0.843 | 0.844 |
| Heart failure | 0.802 | 0.839 | 0.776 |
| Heart rhythm | 0.695 | 0.694 | 0.635 |
| Hip/Knee | 0.853 | 0.814 | 0.849 |
| Kidney | 0.696 | 0.669 | 0.652 |
| Liver | 0.764 | 0.741 | 0.690 |
| Lower back | 0.694 | 0.666 | 0.673 |
| Neck disorder | 0.727 | 0.699 | 0.713 |
| Neuropathy | 0.807 | 0.786 | 0.750 |
| Opioid painkillers | 0.778 | 0.692 | 0.610 |
| Osteoporosis | 0.844 | 0.818 | 0.848 |
| Pacemaker | 0.897 | 0.874 | 0.813 |
| Painkillers | 0.619 | 0.585 | 0.612 |
| Sleep apnea | 0.805 | 0.751 | 0.779 |
| Sleep medication | 0.677 | 0.598 | 0.554 |
| Stroke or TIA | 0.778 | 0.732 | 0.731 |
| Thyroid | 0.760 | 0.728 | 0.769 |
| Urinary | 0.801 | 0.754 | 0.795 |
| Vision | 0.661 | 0.631 | 0.651 |

Table 10: Downstream performance evaluation of PPG and ECG embeddings in predicting age, biological sex and BMI of participants, when the number of pre-training segments in PPG is similar to that for ECG in Table 1.

| Positive pair | Prediction task | PPG | | ECG | |
|---|---|---|---|---|---|
| | | AUC (pAUC) ↑ | MAE ↓ | AUC (pAUC) ↑ | MAE ↓ |
| Participant level (main) | Age classification | 0.974 (0.907) | - | 0.916 (0.763) | - |
| | Age regression | - | 3.38 | - | 6.33 |
| | BMI classification | 0.908 (0.730) | - | 0.797 (0.612) | - |
| | BMI regression | - | 2.54 | - | 3.72 |
| | Sex classification | 0.989 (0.947) | - | 0.951(0.841) | - |

Table 11: Downstream performance evaluation of PPG and ECG embeddings in predicting age, biological sex and BMI of participants, using two types of positive pair selection strategies during pre-training: 1) participant level, 2) segment level, calculated at segment-level granularity, where each segment contributes one and only one sample in the downstream training/evaluation.

| Positive pair | Prediction task | PPG | | ECG | |
|---|---|---|---|---|---|
| | | AUC (pAUC) ↑ | MAE ↓ | AUC (pAUC) ↑ | MAE ↓ |
| Participant level (main) | Age classification | **0.961** (**0.858**) | - | **0.891** (**0.739**) | - |
| | Age regression | - | **4.28** | - | **7.61** |
| | BMI classification | **0.850** (**0.674**) | - | **0.723** (**0.559**) | - |
| | BMI regression | - | **3.33** | - | **4.18** |
| | Sex classification | **0.961** (**0.870**) | - | **0.904** (**0.737**) | - |
| Segment level | Age classification | 0.822 (0.657) | - | 0.748 (0.606) | - |
| | Age regression | - | 8.43 | - | 10.39 |
| | BMI classification | 0.672 (0.555) | - | 0.649 (0.526) | - |
| | BMI regression | - | 4.50 | - | 4.52 |
| | Sex classification | 0.762 (0.592) | - | 0.806 (0.647) | - |

Table 12: Downstream performance evaluation of PPG and ECG embeddings in predicting age, biological sex and BMI of participants, using different pre-training frameworks.

| Pre-training | Prediction task | PPG | | ECG | |
|---|---|---|---|---|---|
| | | AUC (pAUC) ↑ | MAE ↓ | AUC (pAUC) ↑ | MAE ↓ |
| Ours | Age classification | **0.976** (0.907) | - | **0.916** (**0.763**) | - |
| | Age regression | - | **3.19** | - | **6.33** |
| | BMI classification | **0.918** (**0.750**) | - | **0.797** (**0.612**) | - |
| | BMI regression | - | **2.54** | - | **3.72** |
| | Sex classification | **0.993** (**0.967**) | - | **0.951** (**0.841**) | - |
| Ours (no KoLeo) | Age classification | **0.976** (**0.910**) | - | 0.912 (0.756) | - |
| | Age regression | - | 3.22 | - | 6.60 |
| | BMI classification | 0.915 (0.746) | - | **0.797** (**0.612**) | - |
| | BMI regression | - | 2.58 | - | 3.82 |
| | Sex classification | 0.992 (0.962) | - | 0.946 (0.822) | - |
| SimCLR (our variation) | Age classification | 0.975 (0.905) | - | 0.914 (0.759) | - |
| | Age regression | - | 3.22 | - | 6.41 |
| | BMI classification | 0.913 (0.744) | - | 0.796 (**0.612**) | - |
| | BMI regression | - | 2.59 | - | 3.79 |
| | Sex classification | 0.992 (0.961) | - | 0.947 (0.816) | - |
| BYOL (our variation) | Age classification | 0.969 (0.889) | - | 0.900 (0.734) | - |
| | Age regression | - | 3.82 | - | 6.72 |
| | BMI classification | 0.893 (0.721) | - | 0.779 (0.593) | - |
| | BMI regression | - | 2.88 | - | 3.90 |
| | Sex classification | 0.987 (0.946) | - | 0.939 (0.805) | - |

