# OpenReview forum: "Large-scale Training of Foundation Models for Wearable Biosignals"
_ICLR.cc/2024/Conference — ICLR 2024 poster_

### Official Review · Reviewer_EBBD · 2023-10-30

**Soundness:** 3 good
**Presentation:** 3 good
**Contribution:** 3 good
**Rating:** 8
**Confidence:** 5

**Summary:**

The proposed work presents a self-supervised learning (SSL) method for foundation models training using a large, longitudinal, multi-year dataset of unlabelled PPG and ECG samples recorded on Apple Watch devices. The performed experiments showed that the pre-trained models can readily encode participant demographics, conditions and medication. The introduced SSL framework incorporates various techniques, such as stochastic augmentation module or participant level positive pair selection proved to behave better than segment level selection. The work flows logically with a comprehensive analysis of how well PPG and ECG embeddings encode participants' information and ablation study of various parameters used in the work. The motivation of the work is clear and the method was compared to other existing techniques proving its robustness.

**Strengths:**

1) The work proposes pre-training of foundation models to a new domain using real data, acquired in uncontrolled environment, acquired over a  long period of time, what is a good indicator of how robust the method is.
2) Comprehensive analysis that includes comparison with other methods, linear probing to analyze how well both embeddings encode participants' information and a detailed evaluation of which one is more predictive, ablation study including analysis of visual representations after dimensionality reduction, validation loss and dispersion ratio.
3) Presented results support claims made in the work, showing robustness of the introduced method and ability to encode participants' information.
4) Overall the soundness and completeness of the work is good in my opinion.

**Weaknesses:**

1) It would be great to include a figure representing the model showing encoder, MLP head and other implementation details.
2) What was the reason for choosing the InfoNCE loss vs. e.g., the normalized temperature-scaled cross entropy loss (NT-Xent) from SimCLR?
3) Usually large batch sizes and more learning steps are beneficial in SSL, have you experimented with even bigger batch sizes than 256?
4) Could you clarify the reason for selecting this specific encoder and specific embedding sizes, have you experimented with other models?
5) Are you going to make the dataset public?

**Questions:**

1) Would you be able to include results for PPG model trained on a smaller dataset (similar number of segments as ECG)? I believe this would be useful for comparing the behavior of the method given similar amount of data.
2) Have you thought about using both modalities in one model? Do you think that modulating, e.g., ECG with PPG embeddings would improve accuracy? Such approaches are useful when one modality is less descriptive than the other, so I was interested in learning more about your thought of how this would apply to this use case.
3) How easily will it be to improve the current labels given that they were acquired using self-reported metrics and tracing it back seems impossible?

---

> ### Author Response · Authors · 2023-11-19
> **Response to R-EBBD [part 1]**
>
> We thank the reviewer for their kind mention of our manuscript's strengths and pointing out a set of important modifications. We have responded to the reviewer’s comment below, and have addressed them accordingly in the paper. We believe the reviewer’s comments have positively improved our manuscript.
>
> > It would be great to include a figure representing the model showing encoder, MLP head and other implementation details.
>
> We thank the reviewer for pointing this out. We have now added a high-level visualization of our pre-training framework in the **new Appendix Fig. 3**.
>
> > What was the reason for choosing the InfoNCE loss vs. e.g., the normalized temperature-scaled cross entropy loss (NT-Xent) from SimCLR?
>
> This is a good point and we apologize for the confusion. As far as we understand, the term InfoNCE loss and NT-Xent are often used interchangeably for a similar loss function, for example, the SimCLR paper (“*A Simple Framework for Contrastive Learning of Visual Representations*”, Chen et al,., 2020) says “*This loss has been used in previous work (Sohn, 2016; Wu et al., 2018; Oord et al., 2018); for convenience, we term it NT-Xent (the normalized temperature-scaled cross entropy loss)*” and in the CPC paper (“*Representation Learning with Contrastive Predictive Coding*”, van den Oord et al., 2018), the same loss was termed as InfoNCE. To further clarify this, **we have now added NT-Xent term as well to where we define the loss**, “*We use InfoNCE (also known as NT-Xent) to maximize the mutual information*”, and we would like to clarify that the contrastive component of our loss is similar to SimCLR.
>
> > Usually large batch sizes and more learning steps are beneficial in SSL, have you experimented with even bigger batch sizes than 256?
>
> The reviewer raises a very good point. We agree with the reviewer that larger batch sizes have been shown to be beneficial in SSL, for instance in the original SimCLR paper (Chen et al,., 2020). However, several follow up works have shown that adding momentum training and/or regularization and/or different objective functions could alleviate the need for larger batch sizes: examples are BYOL (“*Bootstrap your own latent: A new approach to self-supervised Learning*”, Grill et al., 2020) and Barlow Twins (“*Barlow Twins: Self-Supervised Learning via Redundancy Reduction*”, Zbontar et al., 2021), to name two. More importantly, a recent work has shown that it is possible to even train SimCLR on ImageNet with smaller batch sizes without an important drop in performance (“*Towards Democratizing Joint-Embedding Self-Supervised Learning*”, Bordes et al., 2023). To address this we have now added this to the **new Appendix A.1**: “*Regarding batch size in our pre-training, while early contrastive self-supervised learning works have shown that larger batch size is necessary to improve performance (Chen et al., 2020; 2021), follow up works removed this dependency by changing the pre-training (Grill et al., 2020; Zbontar et al., 2021), and a recent work has shown that it is possible to train SimCLR on ImageNet with smaller batch sizes without an important drop in performance (Bordes et al., 2023). We also did not see meaningful change in performance with larger batch sizes in our early experiments.*”
>
> > Could you clarify the reason for selecting this specific encoder and specific embedding sizes, have you experimented with other models?
>
> We appreciate the reviewer for their careful comment. We picked the EfficientNet model architecture given its efficacy for model performance w.r.t number of parameters. However, to address the reviewer’s comment, we did a **new Ablation 5.2.4** and a **new Table 4** by comparing the performance of our default 1D-EfficientNet model with our variation of 1D-ResNet and 1D-ViT to enable comparison with alternative model architectures and model sizes: “*For comparisons, we used smooth effective rank as a crude proxy of general downstream performance, and observed that different encoder architectures, with different model sizes, can achieve relatively similar performance, demonstrating that the performance is not unique to the 1D-EfficientNet encoder architecture (Table 4). We observed that 1D-EfficientNet model yielded similar performance to these alternative encoders with significantly smaller number of parameters, which was one of the main reasons we picked 1D-EfficientNet as our default backbone encoder given less memory footprint when potentially running on a wearable device. An interesting future direction is scaling up the model size further, particularly with transformer-based models (e.g., 1D-ViT) given their scalability (Kaplan et al., 2020), and larger datasets to study its effect on downstream performance*". Also regarding the embedding size, in our early experiments, we investigated larger and smaller embedding sizes and 256 offered a reasonable sweet point and we did not tune it further afterwards.

---

> > ### Author Response · Authors · 2023-11-19
> > **Response to R-EBBD [part 2]**
> >
> > > Are you going to make the dataset public?
> >
> > This is a very important point. Given the sensitivity of health data and that such data is collected under informed consent, we cannot release the data to general public. However, we have added details for how researchers can request for sourcing the dataset and the code, “*the aggregated data that support the findings of this study can be made available on request from the corresponding author. Request for data will be evaluated and responded to in a manner consistent with the specific language in the study protocol and informed consent form. Similarly, code for all data analyses may be available upon request from the corresponding author. Requests for code will be evaluated and responded to in a manner consistent with policies intended to protect participant confidentiality and language in the study protocol and informed consent form*”, which is now added to the **new Reproducibility statement** of the paper.
> >
> > > Would you be able to include results for PPG model trained on a smaller dataset (similar number of segments as ECG)? I believe this would be useful for comparing the behavior of the method given similar amount of data.
> >
> > We thank the reviewer for their comment about, we agree that this is an important point, and we have now added a **new Appendix Table 10** containing these results.
> >
> > > Have you thought about using both modalities in one model? Do you think that modulating, e.g., ECG with PPG embeddings would improve accuracy? Such approaches are useful when one modality is less descriptive than the other, so I was interested in learning more about your thought of how this would apply to this use case.
> >
> > We thank the reviewer for this comment. We agree that this is an important future direction, and an future work could investigate multi-modal pre-training, similar to CLIP (“*Learning Transferable Visual Models From Natural Language Supervision*”, Radford & Kim et al., 2021) by supervising one modality from the other in pre-training. We think this could improve the downstream performance of weaker modalities by supervising them in the pre-training with a stronger modality, and perhaps let the encoders learn potential relationships between modalities. While this is beyond the scope of our current manuscript, we have now added this point to the **updated Discussion** section as a potential future direction": "*Another future area of investigation of investigation is multi-modal pre-training by: 1) using a multi-modal encoder that takes multiple modalities (e.g., PPG and ECG) as input, 2) or CLIP-style multi-modal pre-training by supervising one modality with another one (Radford et al., 2021), or training multiple encoders for multiple modalities simultaneously (Girdhar et al., 2023), using a contrastive loss*".
> >
> > > How easily will it be to improve the current labels given that they were acquired using self-reported metrics and tracing it back seems impossible?
> >
> > This really depends on the target of interest, but one way to improve upon the self-reported labels is to use labels for standard medical diagnosis such as lab/clinical recordings and standard questionnaires for certain health conditions. This requires collecting such data form informed participants, and the complexity/difficulty could vary from condition to condition.

---

### Official Review · Reviewer_FYDu · 2023-11-01

**Soundness:** 3 good
**Presentation:** 3 good
**Contribution:** 3 good
**Rating:** 5
**Confidence:** 4

**Summary:**

The paper presents a significant advancement in the domain of health monitoring using wearable devices, specifically focusing on the tracking of biosignals, namely photoplethysmography (PPG) and electrocardiogram (ECG). Recognizing the potential of these biosignals, the authors highlight a major challenge: the lack of large, curated datasets with medically annotated labels for developing neural network-based biomarkers. To circumvent this, they leverage the Apple Heart and Movement Study (AHMS) and harness a self-supervised learning framework for training foundation models on PPG and ECG data from approximately 141,207 participants over a span of three years. This self-supervised approach integrates a participant level positive pair selection, stochastic augmentation, and a regularized contrastive loss optimized through momentum training. The authors demonstrate that these pre-trained models encode substantial information related to participant demographics and health conditions. Notably, this work distinguishes itself as the pioneering effort in building foundation models for PPG and ECG using large-scale data sourced from consumer wearables, as opposed to traditionally smaller datasets from clinical settings. The potential applications of these models are vast, with implications for enhancing wearable device capabilities, reducing reliance on labeled data, and ultimately benefiting users' health.

**Strengths:**

Strengths:
1. **First Work on Foundation Models for Wearable Biosignals**:
The research stands out as the pioneering effort to develop foundation models specifically for biosignals—photoplethysmography (PPG) and electrocardiogram (ECG)—collected via wearable devices. Such biosignals offer a treasure trove of biological and cardiac information, which can be instrumental in monitoring users' overall health. The convenience of wearable devices combined with the potential of these foundation models paves the way for continuous health tracking without disrupting daily routines, potentially leading to the early detection of health issues.

2. **Extensive Evaluation Using Linear Probing**:
The authors conducted a comprehensive evaluation of the trained models using linear probing for a plethora of tasks. This includes gender classification, age prediction, and both classification and regression for Body Mass Index (BMI). Furthermore, they delved into predicting variables extracted from questionnaires, providing a holistic understanding of the models' capabilities. The utilization of smooth effective rank as an evaluative measure underscores the robustness of the evaluation.

3. **Participant Level vs. Segment Level Positive Pairs**:
An intriguing facet of the research was the comparison between participant level and segment level positive pairs. This distinction is crucial because the granularity at which positive pairs are chosen can significantly impact the model's performance. The exploration of this dimension offers valuable insights into the optimal approaches for training foundation models on biosignals.

4. **Benchmarking with Other SSL Methods**:
To validate the efficacy of their approach, the authors benchmarked their models against established self-supervised learning (SSL) methods, such as SimCLR and BYOL. Such a comparative analysis not only situates the research within the broader landscape of SSL but also provides tangible metrics to gauge the relative performance of their models.

5. **Curation of a Large Dataset**:
One of the paper's major contributions is the meticulous curation of a vast dataset from the Apple Heart and Movement Study (AHMS) for the training of foundation models. With data spanning approximately 141,207 participants over three years, the effort required to curate, clean, and prepare such a dataset for effective training cannot be understated. This endeavor not only underscores the thoroughness of the research but also sets a precedent for future studies aiming to leverage large-scale datasets for health applications.

**Weaknesses:**

Areas for improvement:

1. **Exploration of KoLeo Regularization**:
An area of potential exploration is the specific impact of the KoLeo regularization on the model's performance. Ablation studies that incrementally remove or vary the strength of KoLeo regularization could provide clarity on its role and efficacy. Such an analysis would help in understanding whether the regularization is crucial for the model's success, and to what extent it contributes to the overall performance. This is particularly important as the unique characteristics of biosignals may interact with regularization techniques differently compared to other domains.

2. **Inclusion of All Results**:
Transparency and completeness in research reporting are essential for reproducibility and peer review. Therefore, it is recommended that the authors include all results, particularly those referenced as "results not shown" within the main body of the paper or the appendix. Having access to these results would allow the scientific community to fully evaluate the findings, methodologies, and claims made within the paper. It would also enhance the credibility and utility of the work for those looking to build upon it.

3. **Augmentation Impact Analysis**:
The paper would benefit from a deeper dive into the effects of various augmentation techniques on the performance of the biosignal models. Unlike image data, where the impact of different augmentations is well-studied, the domain of biosignals remains relatively unexplored in this aspect. An appendix providing detailed results and analysis of how different augmentations influence the learning process would be invaluable. This could include which augmentations contribute most to model robustness or performance, and any augmentation-specific phenomena observed with biosignals. Given the novelty of the field, such insights could be highly influential for future research in biosignal analysis.

4. **Demographic details missing**: The paper would benefit greatly from a more detailed presentation of demographic information related to the participants whose data underpin the foundation models. Such information is essential for assessing the diversity and representativeness of the dataset, which in turn, influences the generalizability of the model across various populations. The current omission of granular demographic details leaves a gap in understanding the scope of the model's applicability. It is recommended that the authors include statistics on age, gender, ethnicity, and other pertinent demographic factors. This would not only enhance the transparency of the research but also allow for a more nuanced evaluation of the model's performance across different demographic groups.

**Questions:**

**Concern Regarding Dataset Availability**:
The dataset from the Apple Heart and Movement Study (AHMS) is central to this research, offering immense value to the broader scientific community. However, the paper doesn't clarify if the dataset will be open-sourced upon acceptance. The release of this dataset, along with the associated models and code, is pivotal for reproducibility and further research.

If the authors don't have ownership of the dataset, detailed instructions on sourcing it would be essential. I kindly request clarity on this matter, as it will influence my final assessment of the paper.

**Details Of Ethics Concerns:**

**Demographic details missing**: The paper would benefit greatly from a more detailed presentation of demographic information related to the participants whose data underpin the foundation models. Such information is essential for assessing the diversity and representativeness of the dataset, which in turn, influences the generalizability of the model across various populations. The current omission of granular demographic details leaves a gap in understanding the scope of the model's applicability. It is recommended that the authors include statistics on age, gender, ethnicity, and other pertinent demographic factors. This would not only enhance the transparency of the research but also allow for a more nuanced evaluation of the model's performance across different demographic groups.

---

> ### Author Response · Authors · 2023-11-19
> **Response to R-FYDu [part 1]**
>
> We appreciate the reviewer for supporting our manuscript, carefully detailing its strengths, and their thoughtful suggestions. We have responded to the reviewer’s comment below, and have addressed them accordingly in the paper. We believe the reviewer’s comments have positively improved our manuscript.
>
> > An area of potential exploration is the specific impact of the KoLeo regularization on the model's performance. Ablation studies that incrementally remove or vary the strength of KoLeo regularization could provide clarity on its role and efficacy. Such an analysis would help in understanding whether the regularization is crucial for the model's success, and to what extent it contributes to the overall performance. This is particularly important as the unique characteristics of biosignals may interact with regularization techniques differently compared to other domains.
>
> We thank the reviewer for pointing this out. We have now performed an experiment where we dropped KoLeo regularization from our pre-training framework to study its importance in isolation. As detailed in the **updated Table 3** and **updated Appendix Table 12**, we observed that removing KoLeo regularization from our pre-training resulted in a drop in performance, evaluated with smooth effective rank and downstream demographic variables’ prediction, for both PPG and ECG. This demonstrates the specific impact of KoLeo regularization and we have now added to the paper: “*We observed that for both PPG and ECG modalities, removing KoLeo regularization from our objective function resulted in reduced evaluation metrics (Table 3 and Appendix Table 12), demonstrating its impact on the model’s performance*”.
>
> > Transparency and completeness in research reporting are essential for reproducibility and peer review. Therefore, it is recommended that the authors include all results, particularly those referenced as "results not shown" within the main body of the paper or the appendix. Having access to these results would allow the scientific community to fully evaluate the findings, methodologies, and claims made within the paper. It would also enhance the credibility and utility of the work for those looking to build upon it.
>
> This is an important point. We have now added the following to the manuscript, providing further information and evidence: 1) **new Appendix Table 10**, including downstream performance evaluation of PPG and ECG embeddings in predicting demographic variables when the number of pre-training segments in PPG is similar to that for ECG in Table 1, 2) **new Appendix Table 11**, including demographic variables’ performance evaluation calculated at segment-level granularity, where each segment contributes one and only one sample in the downstream training/evaluation.
>
> > The paper would benefit from a deeper dive into the effects of various augmentation techniques on the performance of the biosignal models. Unlike image data, where the impact of different augmentations is well-studied, the domain of biosignals remains relatively unexplored in this aspect. An appendix providing detailed results and analysis of how different augmentations influence the learning process would be invaluable. This could include which augmentations contribute most to model robustness or performance, and any augmentation-specific phenomena observed with biosignals. Given the novelty of the field, such insights could be highly influential for future research in biosignal analysis.
>
> We appreciate the reviewer for this thoughtful comment. We have now added details about the selection of our augmentation functions/probabilities in the **new Appendix section A.1** and have done a **new Ablation 5.2.5** with the **new Appendix Fig. 7** regarding the effect of single augmentation functions in our pre-training framework: “*We studied the effect of single augmentation functions for each modality. To do so, for both PPG and ECG, we kept each of the augmentation functions in isolation separately (with probability 1), and repeated our pre-training framework from scratch. We observed that PPG pre-training was more sensitive to the choice of single augmentation functions (more variance in performance), and channel permute and cut out were the most effective augmentation functions in isolation for PPG and ECG, respectively (Appendix Fig. 7). Future work can study more optimal ways to design modality-specific augmentation modules*”.

---

> > ### Author Response · Authors · 2023-11-19
> > **Response to R-FYDu [part 2]**
> >
> > > The paper would benefit greatly from a more detailed presentation of demographic information related to the participants whose data underpin the foundation models. Such information is essential for assessing the diversity and representativeness of the dataset, which in turn, influences the generalizability of the model across various populations. The current omission of granular demographic details leaves a gap in understanding the scope of the model's applicability. It is recommended that the authors include statistics on age, gender, ethnicity, and other pertinent demographic factors. This would not only enhance the transparency of the research but also allow for a more nuanced evaluation of the model's performance across different demographic groups.
> >
> > This is a great suggestion. We have now added the **new Appendix Fig. 5** demonstrating our dataset's (AHMS) demographics (including age, BMI, biological sex, and ethnicity), and we believe it enhances the transparency of our work.
> >
> > > The dataset from the Apple Heart and Movement Study (AHMS) is central to this research, offering immense value to the broader scientific community. However, the paper doesn't clarify if the dataset will be open-sourced upon acceptance. The release of this dataset, along with the associated models and code, is pivotal for reproducibility and further research. If the authors don't have ownership of the dataset, detailed instructions on sourcing it would be essential. I kindly request clarity on this matter, as it will influence my final assessment of the paper.
> >
> > We would like to thank the reviewer for their comment about this. Given the sensitivity of health data and that such data is collected under informed consent, we cannot release the data to general public. However, we have added details for how researchers can request for sourcing the dataset and the code, “*the aggregated data that support the findings of this study can be made available on request from the corresponding author. Request for data will be evaluated and responded to in a manner consistent with the specific language in the study protocol and informed consent form. Similarly, code for all data analyses may be available upon request from the corresponding author. Requests for code will be evaluated and responded to in a manner consistent with policies intended to protect participant confidentiality and language in the study protocol and informed consent form*”, which is now added to the **new Reproducibility statement** of the paper.

---

### Official Review · Reviewer_UQ5r · 2023-11-01

**Soundness:** 3 good
**Presentation:** 4 excellent
**Contribution:** 3 good
**Rating:** 8
**Confidence:** 5

**Summary:**

This paper employs self-supervised learning (SSL) on a large dataset comprising two types of data: photoplethysmography (PPG) and electrocardiogram (ECG) collected from the Apple Watch. The self-supervised learning is carried out on a comprehensive dataset, encompassing a vast span of 141K subjects.
Subsequently, the authors investigate various modules within the SSL framework to glean practical insights. They also elaborate on the effects of certain design choices, such as the selection of positive-negative pairs and data augmentation strategies.
The authors conducted a thorough evaluation and ablation studies for the foundational model they developed. The learned embeddings demonstrate predictive power across a wide array of downstream tasks, such as predicting demographic features and survey questions.

**Strengths:**

- The paper is well-articulated. The authors clearly presented the proposed method, experimental setup, and analysis.

- To the best of my knowledge, this represents the first attempt at training a self-supervised learning (SSL) foundational model for PPG and ECG data on such a grand scale. The results from this process offer valuable scientific insights.

- The authors conducted an exhaustive evaluation of the pretrained models. These pretrained embeddings were assessed against more than 50 diseases.

**Weaknesses:**

- There are potential concerns on the technical methodology front. While this study is the product of extensive training and evaluation, much of the methodology draws from pre-existing studies. Although there are several SSL studies tailored for time-series data, particularly in the realm of biosignals in healthcare, the authors did not extensively compare different model architectures. Readers might be keen to discern whether biosignal SSL performance is more contingent upon scale or the model architecture itself.

- While this study encompasses two modalities, it seems that the authors have considered them in isolation. Pretrained models have shown effectiveness across varied modalities like images and language. It would be beneficial for the authors to delve deeper into this aspect.

- The pretrained embedding for PPG appears to encapsulate more information than its ECG counterpart. This raises a question: given that PPG is passively sampled and ECG is actively collected by users, could this disparity in data collection methods influence such an outcome? Additionally, conventional clinical diagnoses often rely on 12-lead ECG or periodic information like HRV derived from ECG. It would be valuable if the authors could elucidate more why the ECG embedding doesn't seem as informative as the PPG.

- Lastly, the authors note that positive pairs are drawn from the same individual. However, a person's PPG and ECG patterns can vary based on different conditions or circumstances. It might be more insightful for the authors to determine positive pairs by taking additional attributes into account.

**Questions:**

- Even if the authors intend to display only aggregated information, it would be beneficial for them to include a representative visualization of both PPG and ECG signals. This would provide readers, especially those without a healthcare background, with a clearer understanding.

- In Figure 2, both PPG and ECG embeddings demonstrate good separability based on subject IDs. However, I'm curious if two subjects with similar demographic attributes should be distinctly separated.

- While ECG and PPG are pivotal for evaluating physiological status, there are also numerous other parameters to consider, such as HRV, heart rate, and possibly activity levels. The authors might wish to discuss this aspect further.

**Details Of Ethics Concerns:**

No concerns

---

> ### Author Response · Authors · 2023-11-19
> **Response to R-UQ5r [part 1]**
>
> We appreciate the reviewer’s feedback, their kind words about the strengths of our work, and their careful examination of our manuscript. We have responded to the reviewer’s comment below, and have addressed them accordingly in the paper. We believe the reviewer’s comments have positively improved our manuscript.
>
> > There are potential concerns on the technical methodology front. While this study is the product of extensive training and evaluation, much of the methodology draws from pre-existing studies. Although there are several SSL studies tailored for time-series data, particularly in the realm of biosignals in healthcare, the authors did not extensively compare different model architectures. Readers might be keen to discern whether biosignal SSL performance is more contingent upon scale or the model architecture itself.
>
> We thank the reviewer for their thoughtful comment. First, we agree with the reviewer that our study is the product of "*extensive training and evaluation*". We would like to emphasize that “*biosignal SSL*” is largely under-explored at this scale compared to other domains of deep learning and there are no universally-agreed-upon model for wearable biosignals (PPG/ECG) that we can really compare our models to. However, to address the reviewer’s comment, we did a **new Ablation 5.2.4** and **new Table 4** by comparing the performance of our default 1D-EfficientNet model with our variation of 1D-ResNet and 1D-ViT to enable comparison with alternative model architectures and model sizes. As we mentioned in text with different model sizes, different model architectures can reach to reasonable accuracies addressing reviewer’s point about “*SSL performance is more contingent upon scale or the model architecture itself*”. We now also mention that: “*For comparisons, we used smooth effective rank as a crude proxy of general downstream performance, and observed that different encoder architectures, with different model sizes, can achieve relatively similar performance, demonstrating that the performance is not unique to the 1D-EfficientNet encoder architecture (Table 4). We observed that 1D-EfficientNet model yielded similar performance to these alternative encoders with significantly smaller number of parameters, which was one of the main reasons we picked 1D-EfficientNet as our default backbone encoder given less memory footprint when potentially running on a wearable device. An interesting future direction is scaling up the model size further, particularly with transformer-based models (e.g., 1D-ViT) given their scalability (Kaplan et al., 2020), and larger datasets to study its effect on downstream performance*".
>
> > While this study encompasses two modalities, it seems that the authors have considered them in isolation. Pretrained models have shown effectiveness across varied modalities like images and language. It would be beneficial for the authors to delve deeper into this aspect.
>
> This is a great point. We respectfully believe that multi-modal pre-training, similar to CLIP (“*Learning Transferable Visual Models From Natural Language Supervision*”, Radford & Kim et al., 2021) is beyond the scope of our current work, and merits a separate detailed investigation on its own. However, we agree with the reviewer that this is an interesting future direction and we have now added this as a potential future direction in the **updated Discussion** section: "*Another future area of investigation of investigation is multi-modal pre-training by: 1) using a multi-modal encoder that takes multiple modalities (e.g., PPG and ECG) as input, 2) or CLIP-style multi-modal pre-training by supervising one modality with another one (Radford et al., 2021), or training multiple encoders for multiple modalities simultaneously (Girdhar et al., 2023), using a contrastive loss*".

---

> > ### Author Response · Authors · 2023-11-19
> > **Response to R-UQ5r [part 2]**
> >
> > > The pretrained embedding for PPG appears to encapsulate more information than its ECG counterpart. This raises a question: given that PPG is passively sampled and ECG is actively collected by users, could this disparity in data collection methods influence such an outcome? Additionally, conventional clinical diagnoses often rely on 12-lead ECG or periodic information like HRV derived from ECG. It would be valuable if the authors could elucidate more why the ECG embedding doesn't seem as informative as the PPG.
> >
> > We thank the reviewer for their comment. While we agree with the reviewer that ECG signals are conventionally used for diagnosing certain medical conditions, we would like to emphasize that, unlike ECG, PPG is relatively under-explored in the medical field and harder to annotate (e.g., it's easy to visually inspect ECG based on the P-QRS-T waveform, however, there are limited gold-standard manual/visual procedures like that for PPG). As we had xplained in the Results section, we believe there are likely two reasons for PPG signals being better than ECG: “*ECG signals may contain less information specific to these conditions, and/or our pre-training framework is more optimal for PPG compared to ECG (see Ablation 5.2.2 and Discussion)*” and had mentioned in the Discussion section: "*Our pre-training framework generalizes to PPG and ECG biosignals, however, we believe future work could gain improvements by incorporating modality-specific design choices*". Related to this, we have now added a new Appendix Table 10 showing that PPG still outperforms ECG with similar number of pre-training segments, shedding light on the fact that PPG embeddings being more informative than ECG embeddings is not due to the number of pre-training segments. Also, we do not believe that the data collection disparity (active vs. passive) could cause this amount of information gap between PPG/ECG, but testing that requires some infrastructural changes to the Apple watch and how these biosignals are recorded, which is beyond the scope of our work.
> >
> > > Lastly, the authors note that positive pairs are drawn from the same individual. However, a person's PPG and ECG patterns can vary based on different conditions or circumstances. It might be more insightful for the authors to determine positive pairs by taking additional attributes into account.
> >
> > We appreciate the reviewer’s comment regarding this. We totally agree with the reviewer that different positive pair selection strategies should be thoroughly examined and could likely improve the models for particular tasks. We respectfully believe this investigation is beyond the scope of our current study, however, to address the reviewer's comment, we have now added this as a future direction in the **updated Discussion** section: “*Last but not least, we observed that the choice of positive pairs significantly affects the quality of the embeddings; future work can investigate the efficacy of different positive pair selection strategies on different health-related targets, by accounting for temporal and other physiological information*”.
> >
> > > It would be beneficial for them to include a representative visualization of both PPG and ECG signals. This would provide readers, especially those without a healthcare background, with a clearer understanding.
> >
> > This is a great suggestion and we have now added representative examples of PPG/ECG processed signals in the **new Appendix Fig. 6**.

---

> > > ### Author Response · Authors · 2023-11-19
> > > **Response to R-UQ5r [part 3]**
> > >
> > > > In Figure 2, both PPG and ECG embeddings demonstrate good separability based on subject IDs. However, I'm curious if two subjects with similar demographic attributes should be distinctly separated.
> > >
> > > This is a good question. Given that our models are trained on unlabeled data, there are various factors that could determine the information encoded in the embeddings from PPG/ECG beyond only demographics (that is evident from Fig. 1a and we know that PPG/ECG embeddings contain more information than just demographics and measured heart rate). In addition to that, T-SNE is a nonlinear dimensionality reduction (Maaten & Hinton, 2008), and the local distances in T-SNE dimensions (what is visually depicted in Fig. 2a) do not linearly represent the global distances. Furthermore, based on the decent predictive power of PPG/ECG embedding about demographics, we know that demographics are encoded well in the embedding subspace (256-D). Given these three factors, we believe that the participants who have similar demographic variables may be close in the embedding subspace (at least in some feature dimension, as evident from decent demographics prediction performance), but may not necessarily lie on top of each other in the T-SNE subspace given other information content in PPG/ECG and that T-SNE would still non-linearly separate them if they’re not exactly on top of each other in the 256-D embedding subspace.
> > >
> > > > While ECG and PPG are pivotal for evaluating physiological status, there are also numerous other parameters to consider, such as HRV, heart rate, and possibly activity levels. The authors might wish to discuss this aspect further.
> > >
> > > We totally agree with the reviewer and believe that future work can investigate the relationship between health conditions, PPG/ECG embeddings, and other factors such as HRV/HR/activity levels. We also believe that metrics such as HR/HRV/activity levels, still provide valuable information regarding one’s health status and should not be neglected. To address this, we have added to the **updated Discussion** section: “*Also, we would like to note that while PPG/ECG embeddings are predictive of health conditions, other biomarkers including but not limited to heart rate, heart rate variability (HRV), and physical activity still provide valuable insight regarding one's health status, and their relationship to PPG and ECG embeddings should be examined further in future work*”.

---

### Official Review · Reviewer_4W8A · 2023-11-04

**Soundness:** 3 good
**Presentation:** 3 good
**Contribution:** 3 good
**Rating:** 6
**Confidence:** 5

**Summary:**

This paper introduced foundation models for wearable sensor data using a large-scale population dataset of ECG and PPG signals. Its embeddings demonstrate generalization across an array of downstream tasks related to personalization and health inferences.

**Strengths:**

- Important domain with very limited available models
- Solid results across an impressive array of downstream tasks
- Careful tuning and experimentation considering the idiosyncrasies of the data

**Weaknesses:**

- Some missing references and links to previous works
- Lack of discussion around scaling up the proposed models
- No discussion around model/data release

**Questions:**

- The paper could better address previous research. For example, it claims that previous works employed biosignals recorded in clinical or controlled experimental settings, however, both [1] and [2] used large-scale _free-living_ datasets in the wild.

- Regarding the augmentations, it is not clear whether assigned probabilities are found through hyper-parameter tuning or heuristics. I point the authors to this paper for further experimental decisions about the order and impact of these augmentations [3].

- There are very limited details about the availability of the dataset. Should we just assume it is private? Are there any public datasets that we could replicate (some of) the results of the paper? I would appreciate any discussion around these points.

- Given the number of participants, the paper could also attempt to increase the sequence length of the signals and assess whether longitudinal/day-level temporal dynamics can impact downstream tasks.

- The linear probing results and performance analysis should be put in context to previous works like [4], [2], and [5].

- The parameter size of the final model seems quite low considering the dataset size and number of participants. The paper could justify the word "foundation model" in its title by scaling up the experiments. For example, it should have been very exciting to find the Chinchilla-optimal parameter size for this sort of data. I would appreciate any discussions here around this topic, are there overfitting issues with bigger models? What about different architectures like Transformers?

[1] Yuan, H., Chan, S., Creagh, A. P., Tong, C., Clifton, D. A., & Doherty, A. (2022). Self-supervised learning for human activity recognition using 700,000 person-days of wearable data. arXiv preprint arXiv:2206.02909.

[2] Spathis, D., Perez-Pozuelo, I., Brage, S., Wareham, N. J., & Mascolo, C. (2021, April). Self-supervised transfer learning of physiological representations from free-living wearable data. In Proceedings of the Conference on Health, Inference, and Learning (pp. 69-78).

[3] Tang, C. I., Perez-Pozuelo, I., Spathis, D., & Mascolo, C. (2020). Exploring contrastive learning in human activity recognition for healthcare. arXiv preprint arXiv:2011.11542.

[4] Wu, X., Huang, C., Robles-Granda, P., & Chawla, N. V. (2022). Representation learning on variable length and incomplete wearable-sensory time series. ACM Transactions on Intelligent Systems and Technology (TIST), 13(6), 1-21.

[5] Hallgrímsson, H. T., Jankovic, F., Althoff, T., & Foschini, L. (2018). Learning individualized cardiovascular responses from large-scale wearable sensors data. arXiv preprint arXiv:1812.01696.

---

> ### Author Response · Authors · 2023-11-19
> **Response to R-4W8A [part 1]**
>
> We appreciate the reviewer’s feedback, their kind words about the strengths of our work, and their careful examination of our manuscript. We have responded to the reviewer’s comments below, and have addressed them accordingly in the paper. We believe the reviewer’s comments have positively improved our manuscript.
>
> > The paper could better address previous research. For example, it claims that previous works employed biosignals recorded in clinical or controlled experimental settings, however, both [1] and [2] used large-scale free-living datasets in the wild.
>
> We thank the reviewer for pointing out an improvement regarding this. We would like to clarify that throughout the manuscript, we had meant to claim that our manuscript is the first study on “*large-scale PPG and ECG data*”, however, we agree with the reviewer that some of the mentioned work could have been better discussed in our paper, and **we have made improvements to the Introduction and Related work sections** to address the reviewer’s comment about the prior work and to include them. The changes for this comment are:
>
> - We modified the Introduction section: “*Self-supervised learning often does not require explicit labels, making it suitable to pre-train foundation models on unlabeled biosignals, and has been recently proven successful for health applications (Hallgrimsson et al., 2018; Cheng et al., 2020; Kostas et al., 2021; Sarkar & Etemad, 2022; Mohsenvand et al., 2020; Gopal et al., 2021; Kiyasseh et al., 2021; Mehari & Strodthoff, 2022; Wu et al., 2020; Spathis et al., 2021; Tang et al., 2021; Yuan et al., 2023; Lai et al., 2023). While there have been recent efforts to apply SSL on wearable accelerometer data in free-living conditions (Spathis et al., 2021; Yuan et al., 2023), other SSL work have mostly used biosignal modalities such as PPG, ECG, and electroencephalogram (EEG) collected in clinical or controlled experimental settings (Cheng et al., 2020; Kostas et al., 2021; Sarkar & Etemad, 2022; Mohsenvand et al., 2020; Gopal et al., 2021; Kiyasseh et al., 2021; Mehari & Strodthoff, 2022; Lai et al., 2023)*“.
>
> - We modified the Related work section: “*Recent work have used self-supervised learning for wearable accelerometer signals in free-living conditions and large dataset (Spathis et al., 2021; Yuan et al., 2023)*”.
>
> - We would like to respectfully clarify that we still believe our claim in Abstract “*To the best of our knowledge, this is the first study that builds foundation models using large-scale PPG and ECG data collected via wearable consumer devices — prior works have commonly used smaller-size datasets collected in clinical and experimental settings.*” is still accurate in consideration of the works the reviewer has mentioned as they use accelerometer data, which is a different modality compared to PPG/ECG.
>
> > Regarding the augmentations, it is not clear whether assigned probabilities are found through hyper-parameter tuning or heuristics. I point the authors to this paper for further experimental decisions about the order and impact of these augmentations [3].
>
> We thank the reviewer for this careful comment. We have now added details about the selection of our augmentation functions/probabilities in the **new Appendix section A.1** and have done a **new Ablation 5.2.5** with **new Appendix Fig. 7** regarding the effect of single augmentation functions in our pre-training framework. We now mention that “*we did not comprehensively tune our augmentation module probabilities, we started with a preliminary set of probabilities in our early experiments based on prior work (Cheng et al., 2020; Tang et al., 2021), for instance, cut out was shown to be the most effective augmentation in (Cheng et al., 2020) and we assigned a higher probability to it. We did not tune these probabilities afterwards and the only change made was to make the ECG augmentation probabilities stronger (2x probability) to allow for more mismatch between positive pair representations*” and that “*we studied the effect of single augmentation functions for each modality. To do so, for both PPG and ECG, we kept each of the augmentation functions in isolation separately (with probability 1), and repeated our pre-training framework from scratch. We observed that PPG pre-training was more sensitive to the choice of single augmentation functions (more variance in performance), and channel permute and cut out were the most effective augmentation functions in isolation for PPG and ECG, respectively (Appendix Fig. 7). Future work can study more optimal ways to design modality-specific augmentation modules*”.

---

> > ### Author Response · Authors · 2023-11-19
> > **Response to R-4W8A [part 2]**
> >
> > > There are very limited details about the availability of the dataset. Should we just assume it is private? Are there any public datasets that we could replicate (some of) the results of the paper? I would appreciate any discussion around these points.
> >
> > We thank the reviewer about their thoughtful comment. Given the sensitivity of health data and that such data is collected under informed consent from participants, we cannot release the data to general public. Also, we are not aware of any public dataset that contains PPG/ECG recorded from wearable devices at this scale, which is the cornerstone of our manuscript, results and evaluations, that we respectfully believe is valuable for the researchers. However, to guide the researchers on how to source the data and the code, we have now added a **new Reproducibility statement**: “*the aggregated data that support the findings of this study can be made available on request from the corresponding author. Request for data will be evaluated and responded to in a manner consistent with the specific language in the study protocol and informed consent form. Similarly, code for all data analyses may be available upon request from the corresponding author. Requests for code will be evaluated and responded to in a manner consistent with policies intended to protect participant confidentiality and language in the study protocol and informed consent form*”.
> >
> >
> > > Given the number of participants, the paper could also attempt to increase the sequence length of the signals and assess whether longitudinal/day-level temporal dynamics can impact downstream tasks.
> >
> > We thank the reviewer for this comment. While this is a good suggestion, we respectfully believe this is beyond the scope of our current manuscript because the PPG/ECG data in our dataset is not sampled regularly and this is not immediately testable; PPG is collected intermittently and passively in Apple Watch and ECG is collected when initiated by the participants, as explained in 4.1. However, we think exploring longitudinal variations in PPG/ECG embeddings is a valuable future direction so we have now mentioned in the **updated Discussion section**: "*Future work can also investigate the longitudinal changes in PPG and ECG embeddings, and whether accounting for those can improve downstream predictions*".
> >
> > > The linear probing results and performance analysis should be put in context to previous works like [4], [2], and [5].
> >
> > This is a great point and we thank the reviewer for pointing us to the related work. We have now added a detailed discussion regarding this in the **new Appendix section A.4** and mentioned that in the main Discussion: “*Prior work have looked into predicting certain demographic variables via representation learning from biosignals: 1) Using activity, sleep, and heart rate, BMI was predicted with ROC AUC of 0.697 (with the binary class cut-off at 30* $\text{kg}/\text{m}^2$ *similar to us), which is significantly smaller than that with our PPG/ECG embeddings in Table 2. Also, age was predicted with ROC AUC of 0.701 (with the binary class cut-off at 31 years) which is not directly comparable to our evaluation due to different cut-offs (Hallgrimsson et al., 2018). 2) A prior work pre-trained models using wearable accelerometer and heart rate data in free-living conditions (Spathis et al., 2021), and predicted age, BMI, and biological sex with ROC AUC of 0.676, 0.694, and 0.934, respectively, where the binary class cut-off was set to the median for continuous variables such as age and BMI. While the biological sex prediction is more accurately predicted with PPG and ECG embeddings in Table 2, the age and BMI predictions are not directly comparable due to different factors including different cut-off for binary labels and different dataset distribution, e.g., the distribution of the age in AHMS (Fig. 5) is different from the dataset used in this prior work (age ranging from 35 to 65). 3) Using the heart rate time-series data, biological sex was predicted with 0.703 and 0.672 Micro and Macro F1 score (Wu et al., 2020) - these reported numbers are lower than that for PPG embeddings (0.973 and 0.969, respectively) and for ECG embeddings (0.892 and 0.869, respectively). This prior work also reports age prediction which is not directly comparable to us due to different class buckets*”.

---

> > > ### Author Response · Authors · 2023-11-19
> > > **Response to R-4W8A [part 3]**
> > >
> > > > The parameter size of the final model seems quite low considering the dataset size and number of participants. The paper could justify the word foundation model in its title by scaling up the experiments. For example, it should have been very exciting to find the Chinchilla-optimal parameter size for this sort of data. I would appreciate any discussions here around this topic, are there overfitting issues with bigger models? What about different architectures like Transformers?
> > >
> > > We thank the reviewer for pointing this out. To better address the reviewer’s comment, we did a **new Ablation 5.2.4** and **new Table 4** by comparing the performance of our default 1D-EfficientNet model with our variation of 1D-ResNet and 1D-ViT to enable comparison with alternative model architectures and model sizes: “*For comparisons, we used smooth effective rank as a crude proxy of general downstream performance, and observed that different encoder architectures, with different model sizes, can achieve relatively similar performance, demonstrating that the performance is not unique to the 1D-EfficientNet encoder architecture (Table 4). We observed that 1D-EfficientNet model yielded similar performance to these alternative encoders with significantly smaller number of parameters, which was one of the main reasons we picked 1D-EfficientNet as our default backbone encoder given less memory footprint when potentially running on a wearable device”. We respectfully believe further analyses on this front is beyond the scope of our current manuscript, and therefore we have framed it as a future direction: “An interesting future direction is scaling up the model size further particularly with transformer-based models (e.g., 1D-ViT given their scalability (Kaplan et al., 2020), and larger datasets to study its effect on downstream performance*".

---

### Author Response · Authors · 2023-11-19
**Summary of the revision and response to the reviewers**

We would like to thank the reviewers for supporting our manuscript and their thoughtful feedback. Overall the reviewers all agreed with the strengths of our work with “*solid results across and impressive array of downstream tasks*” in an “*important domain with very limited available models*” **[R-4W8A]**, that our manuscript “*is well-articulated*” and “*clearly presented the proposed method, experimental setup and analysis*” **[R-UQ5r]** with “*extensive evaluation*” and “*benchmarking*” **[R-FYDu]**, and they mentioned that “*the soundness and completeness of the work is good in my opinion*” **[R-EBBD]**. We also thank the reviewers for their constructive comments. We have incorporated the reviewers’ suggestions, and made the following changes, which we think have significantly improved the quality of our manuscript:

- **Ablation study for the encoder architecture and model size**: we performed an ablation study to investigate the efficacy of different model architectures (1D-EfficientNet, 1D-ResNet and 1D-ViT) with different model sizes (up to 16.9M number of parameters), and to show that the performance is not unique to the model architecture. This analysis is added to the new main text Ablation 5.2.4 and the new Table 4.
- **Ablation study for augmentation functions**: we studied the effect of different augmentation functions for PPG and ECG pre-training in the new main text Ablation 5.2.5 and the new Appendix Fig. 7.
- **Studying the effect of KoLeo regularization**: we studied the effect of KoLeo regularization for both PPG and ECG in our pre-training framework in updated main text Ablation 5.2.3, updated Table 3, and updated Appendix Table 12.
- **Comparison of PPG/ECG embeddings with the same number of segments in the pre-training dataset**: We added a control analysis for the comparison of PPG and ECG in predicting downstream demographic variables, when the number of pre-training segments in PPG dataset is similar to ECG. The results are added to the new Appendix Table 10.
- **Improved discussion about prior work**: We have improved the discussions about the prior work by updating the Introduction, Related work sections and the new Appendix section A.4.
- **Improved discussion about batch size and augmentation choices for our pre-training**: we added discussion about the choices of batch size and augmentation functions in our pre-training in the new Appendix section A.1.
- **Multiple discussion items about future work**: we added multiple future directions in the updated Discussion section and throughout the text about multi-modal pre-training, different positive pair selection strategies, longitudinal variations in PPG/ECG embeddings, taking into account other physiological parameters, studying scaled up dataset and model size, and modality specific augmentation functions.
- **Multiple new figures about the pre-training framework and dataset**: we added multiple new figures for our pre-training framework visualization (new Appendix Fig. 3), the distribution of demographic variables in AHMS (new Appendix Fig. 5), example PPG/ECG segments (new Appendix Fig. 6).
- **Reproducibility statement**: we added a "Reproducibility statement" to our manuscript, conformed with the AHMS study protocol and informed consent form from participants. The reproducibility statement explains how researchers can request for sourcing the dataset and the code upon acceptance.
- **Modified the organization of the manuscript**: we slightly modified the organization of the manuscript to fit it within the 9-page limit while incorporating the new changes.

---

### Meta-Review · Area_Chair_PtTB · 2023-12-06

**Metareview:**

This paper presents novel foundation models for understanding PPG (photoplethysmograph - pulse ox) and ECG (electrocardiograph) data collected via wearable consumer devices using extremely large scale data.  This is an important contribution to a domainwhere analysis is often restricted to very limited available models.  The authors demonstrate solid results across an impressive array of downstream tasks
Careful tuning and experimentation considering the idiosyncrasies of the data


The main strengths are:

-The problem is important
-This appears to be the first model developed of its type
-The experiments seem to be well conducted and explained

Weaknesses have been substantially addressed by the authors

**Justification For Why Not Higher Score:**

This work is interesting and well executed but it is likely of high interest to only a subset of the ICLR community

**Justification For Why Not Lower Score:**

There seems to be no reason to reject this paper.

---

### Decision · Program_Chairs · 2024-01-16

Accept (poster)